# Seasonal modulation of phytoplankton biomass in the Southern Ocean

Lionel A. Arteaga [1,2✉], Emmanuel Boss[3], Michael J. Behrenfeld[4], Toby K. Westberry [4] & Jorge L. Sarmiento[1]

Over the last ten years, satellite and geographically constrained in situ observations largely focused on the northern hemisphere have suggested that annual phytoplankton biomass cycles cannot be fully understood from environmental properties controlling phytoplankton division rates (e.g., nutrients and light), as they omit the role of ecological and environmental loss processes (e.g., grazing, viruses, sinking). Here, we use multi-year observations from a very large array of robotic drifting floats in the Southern Ocean to determine key factors governing phytoplankton biomass dynamics over the annual cycle. Our analysis reveals seasonal phytoplankton accumulation ('blooming') events occurring during periods of declining modeled division rates, an observation that highlights the importance of loss processes in dictating the evolution of the seasonal cycle in biomass. In the open Southern Ocean, the spring bloom magnitude is found to be greatest in areas with high dissolved iron concentrations, consistent with iron being a well-established primary limiting nutrient in this region. Under ice observations show that biomass starts increasing in early winter, well before sea ice begins to retreat. The average theoretical sensitivity of the Southern Ocean to potential changes in seasonal nutrient and light availability suggests that a 10% change in phytoplankton division rate may be associated with a 50% reduction in mean bloom magnitude and annual primary productivity, assuming simple changes in the seasonal magnitude of phytoplankton division rates. Overall, our results highlight the importance of quantifying and accounting for both division and loss processes when modeling future changes in phytoplankton biomass cycles.

[1] Program in Atmospheric and Oceanic Sciences, Princeton University, 300 Forrestal Rd, Princeton, NJ, USA. [2] NASA Global Modeling and Assimilation Office, Universities Space Research Association, Greenbelt, MD 20771, USA. [3] School of Marine Sciences, University of Maine, 5706 Aubert Hall, Orono, ME 04469-5741, USA. [4] Department of Botany and Plant Pathology, Oregon State University, Cordley Hall 2082, Corvallis, OR 97331-2902, USA. ✉email: lionel.arteagaquintero@nasa.gov

The photosynthetic production of organic carbon by marine phytoplankton plays a key role in regulating atmospheric carbon dioxide ($CO_2$) levels, such that without this biological uptake it is estimated that present day atmospheric $CO_2$ concentrations would be 200 ppm (50%) higher[1]. Phytoplankton blooms in the temperate and polar oceans play a disproportionally large role in ocean $CO_2$ uptake, as well as being critical ecological events to which the migration patterns of marine animals, ranging from zooplankton to whales, have evolved[2]. The cause of phytoplankton blooms has traditionally been attributed to seasonal changes in 'bottom-up' environmental factors controlling phytoplankton division rates, such as nutrients and light[3–7]. However, seasonal changes in phytoplankton biomass ($P$) represented by the biomass-specific net rate of change ($r$) always reflect the interplay between two dominant terms, the phytoplankton division rate ($\mu$) and the sum of all loss ($l$) rates (e.g., grazing, viruses, sinking):

$$r = \frac{1}{P}\frac{dP}{dt} = \mu - l, \qquad (1)$$

implying that a 'bottom-up' interpretation of blooms is, by necessity, incomplete[8–10]. The importance of seasonal variations in loss rates has recently been highlighted by satellite and in situ studies demonstrating that annual blooming events often begin in early winter when phytoplankton division rates are still declining[10–15], but these earlier investigations have largely focused on regions of the northern hemisphere. Here, we use multi-year in situ bio-optical measurements from 146 robotic drifting floats in the Southern Ocean (south of 30°S), in conjunction with satellite data, to resolve ecological drivers of phytoplankton biomass cycles. Our results demonstrate a closely coupled interplay between 'bottom-up' and 'top-down' (i.e., loss) processes controlling the onset and temporal evolution of Southern Ocean blooms. These 'bloom-forming' mechanisms have been previously summarized by the 'Disturbance-Recovery' hypothesis based predominantly on satellite observations[11,15]. Our results provide large scale observational evidence in support of this hypothesis, based primarily on in situ biogeochemical and bio-optical data from autonomous drifting floats. Integrating this in situ-based finding into a productivity model indicates that small changes in phytoplankton division rates associated with predicted changes in Southern Ocean environmental conditions may result in disproportionately large decreases in future bloom magnitude and primary production.

## Results

### Biomass cycles in the Southern Ocean.
For the current analysis, we used float measurements collected between 6 March 2012 and 12 March 2019, which provided broad coverage of the Southern Ocean region (Fig. S1). Physical as well as biotic (biomass and growth-related) variables were initially obtained for each individual float profile (Fig. S2) and subsequently averaged into different zones, highlighting diverse environmental conditions of the Southern Ocean. Phytoplankton blooms are here defined as large-scale regionally averaged periods of positive of net rate of change of phytoplankton biomass ($r > 0$), which differs from short time-scale blooming events that are often observed at small spatial scales in the field. Annual cycles of phytoplankton biomass were obtained from empirical relationships between float-measured particulate backscatter coefficients at 700 nm ($b_{bp}(700)$) and phytoplankton carbon (Methods). These data show that average phytoplankton biomass for the Southern Ocean as a whole is highest (~900 mg C m$^{-2}$) during austral summer (January–February) (Fig. 1) and exhibits a seasonal cycle correlated with the shoaling and deepening of the mixed layer, the average light level within the mixed layer, and seasonal changes in phytoplankton

division rates (Methods). Interestingly, mixed layer-averaged modeled phytoplankton division rates ($\mu$) are about 2–3 months time-lagged behind the net biomass rate of change ($r$), suggesting that seasonal changes in biomass are not exclusively driven by 'bottom-up' factors. Moreover, values of $r$ are ~100 times lower than $\mu$, indicating that growth and loss processes must be tightly coupled and of similar magnitude. The validity of modeled division rates was assessed by comparing them with a productivity algorithm parameterized specifically for Southern Ocean waters[16] as well as division rate estimates inferred from in situ carbon-14 ($^{14}$C) based net primary productivity measurements[17] (Supplementary Information, section 'Assessment of division rates estimated by the CbPM').

Initiation of the blooming period (BI, Fig. 1c) can be identified by a negative-to-positive change in sign of the net biomass rate of change, $r$. In the four annual cycles of biomass observed between 2015 and 2019, BI occurs at the end of winter when incident sunlight is close to lowest, phytoplankton division rates are near-minimal, and mixing is deepest. Also counterintuitively, bloom termination (BT), marked by a positive-to-negative sign change in $r$, occurs when phytoplankton division rates are near-maximal. The temporal misalignment between the division rate ($\mu$) and the net biomass rate of change ($r$) can only be explained by subtle seasonal changes in the balance between $\mu$ and loss ($l$) rates.

Additional insight on processes affecting seasonal cycles of biomass is provided by changes in the rate of change (slope) of $r$. The moment when $r$ stops decreasing (but is still <0) marks the time in autumn when the net loss of biomass is highest (minimum $r$, $rM$) (Fig. 1). This event begins in early winter while conditions for phytoplankton growth are still deteriorating, but the rate of decrease in $\mu$ begins to slow (Fig. 1c). These findings show that the net phytoplankton biomass rate of change ($r$) does not covary with the absolute value of $\mu$, but rather with the rate of change in $\mu$ (Fig. 2). Such a relationship will exist when division and loss rates are tightly coupled, but a temporal lag exists in the response time for the loss processes[10,15,18].

Even when integrated over our full Southern Ocean domain, the extensive float record analyzed here immediately highlights the important role of predator-prey relationships in terms of governing the annual phytoplankton biomass cycle. The temporal misalignment between $\mu$ and $r$ can also be observed in individual float time series (Fig. S4). The synoptic view of the float-based multi-year record (Fig. 1) agrees with the broad-scale dynamics of upper ocean planktonic ecosystems inferred from satellite observations for large regions of the high latitude ocean[15]. However, the Southern Ocean is comprised of well-established distinct environmental zones that can provide more detailed understanding of biomass variability in this large region of the global ocean (Fig. S1). We therefore subdivided the Southern Ocean into four primary zones of differing physical and biogeochemical characteristics (Methods): a Subtropical Zone (STZ) roughly encompassing oligotrophic waters between 30°S and 40°S, a Subantarctic Zone (SAZ) and a Polar Antarctic Zone (PAZ) that together cover the circumpolar section between approximately 40°S and 60°S, and a Seasonal Ice Zone (SIZ) representing seasonally ice-covered areas between Antarctica and ~60°S. For each zone, we evaluated seasonal patterns in phytoplankton biomass to identify key mechanisms driving variations in the net biomass rate of change.

### Subantarctic and Polar Antarctic Zones.
The SAZ and PAZ show similar annual cycles of $r$, with bloom initiation (at the beginning of the blooming phase) occurring in July and corresponding to near-minimal phytoplankton division rates (Fig. 2). As observed for the integrated Southern Ocean (Fig. 1), peak

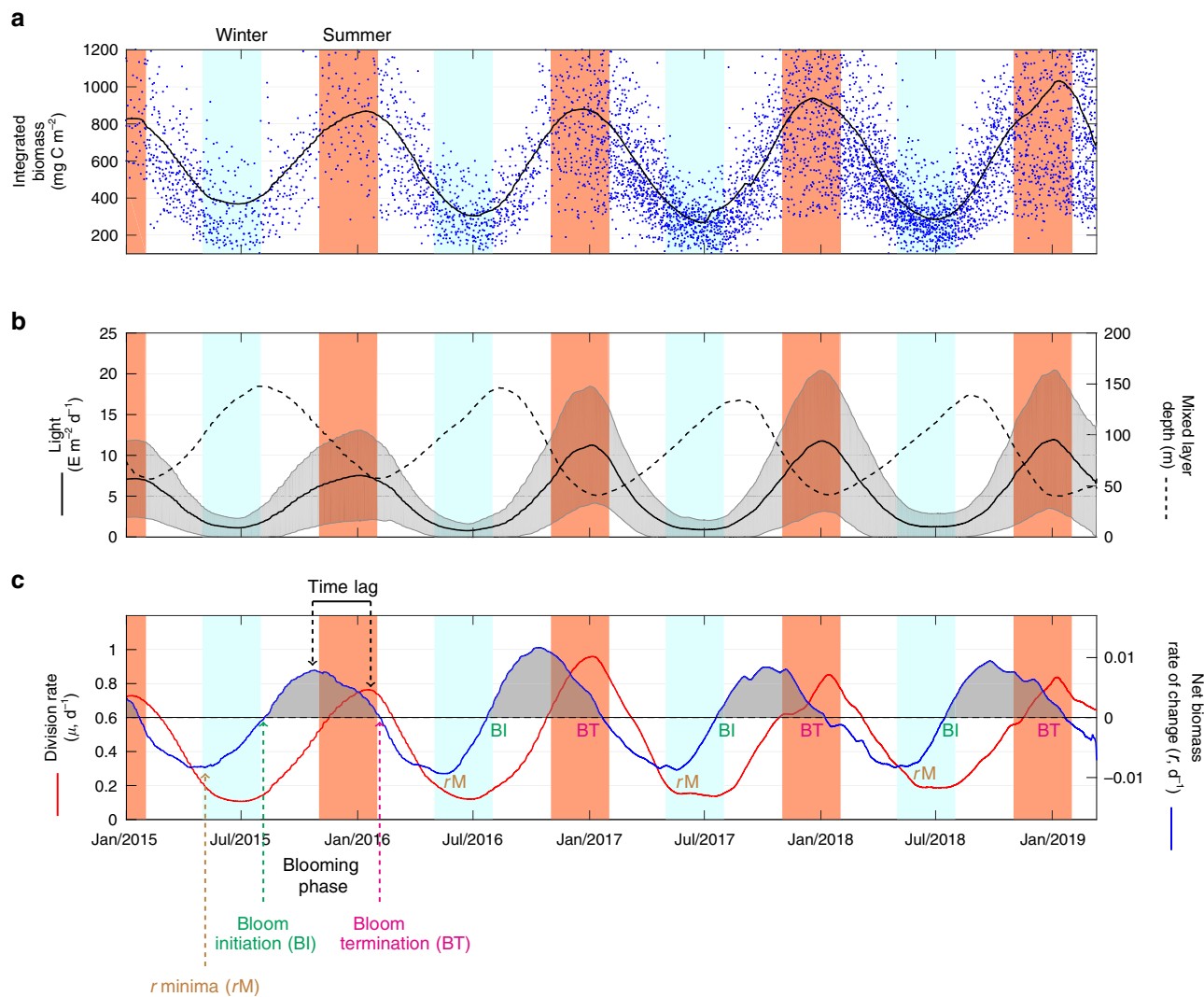

**Fig. 1 Annual cycles of phytoplankton biomass in the Southern Ocean. a** Annual cycles of phytoplankton carbon integrated from surface to the depth of the mixed layer or euphotic depth (whichever is deeper). Blue dots: Individual float observations. Continuous black line: Average time series of phytoplankton carbon from individual float-based observations for the Southern Ocean. **b** Black continuous line: Average time series of the mean light level in the surface mixed layer in the Southern Ocean computed as photosynthetically active radiation (PAR) (shaded area represents the standard deviation). Black dashed line: Average time series of the depth of the surface mixed layer. **c** Red continuous line: Average time series of phytoplankton division rates ($\mu$). Blue continuous line: Average time series of phytoplankton net biomass rate of change rate ($r$). The phytoplankton blooming phase is defined as the time period where $r > 0$ (gray shaded area in (**c**)), constrained between the time of `Bloom initiation' (BI) and `Bloom termination' (BT) of each annual cycle. The `$r$ minima' ($r$M) point, indicates the moment in which autumn biomass starts to decline at a lower rate prior to the onset of the bloom. Seasonally, a clear `Time lag' exists between $\mu$ and $r$ where the highest net biomass rates of change are observed ~3 months before the peak in division rates. Light blue and red shaded panels indicate austral winter (May–August) and summer (November–February) months, respectively. Averaged time series (**a**–**c**) are based on individual float-based observations for the Southern Ocean. See "Methods" for details on the smoothing of time series.

values of $\mu$ for the SAZ and PAZ occur ~3 months after the annual peak in net biomass rate of change ($r$). In contrast, the annual cycle in $r$ is temporally aligned with that of the division rate of change ($d\mu/dt$, i.e., the temporal derivative of $\mu$). Satellite observations of the polar zones earlier revealed $d\mu/dt$ as a principal driver of variation in phytoplankton concentration[15]. The interpretation of this finding has been that accelerations in $\mu$ (that is, $d\mu/dt > 0$) result in an accumulation of biomass because they allow phytoplankton division to outpace growing loss rates, whereas decelerations in $\mu$ ($d\mu/dt < 0$) result in increased loss from overgrazing and thus declining biomass. In this view of annual phytoplankton cycles, the importance of 'bottom-up' factors resides in their influence on 'top-down' predator-prey relations and, for the Southern Ocean, plays out in synchrony with seasonal changes mixed layer light levels (Fig. 1).

In addition to the dominant spring bloom, the SAZ also exhibits a less-pronounced autumn bloom that corresponds to the initial deepening of the mixed layer. One potential explanation for this feature is that it reflects an entrainment of deeper phytoplankton populations into the mixed layer, but analysis of our float time-series data rarely showed the enhanced deep-water biomass prior to mixed layer deepening that would be necessary to support this explanation. Alternatively, autumn mixing could be envisioned to enhance mixed layer nutrient concentrations and thus stimulate blooming, but this interpretation is not supported by estimated division rates during this period (Fig. 2), noting however that our phytoplankton growth model does not explicitly resolve unique attributes of iron stress (Methods)[19]. A direct - 'physical trigger' on grazing rates for the SAZ autumn blooms may be the primary driver of this event, where deepening

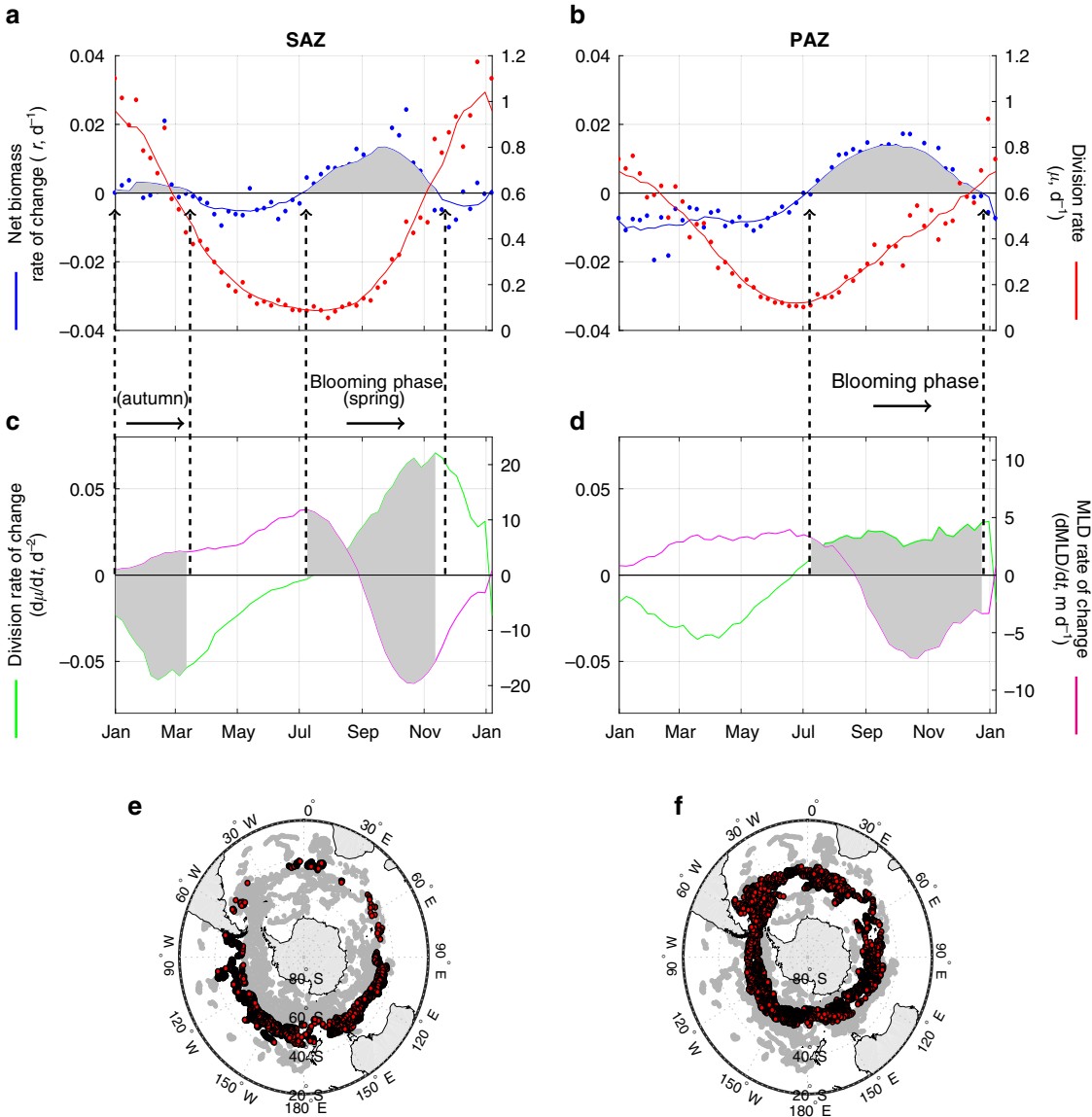

**Fig. 2 Climatological bloom cycles in the Subantarctic and Polar Antarctic Zone (SAZ and PAZ). a**, **b** The annual cycle of net phytoplankton biomass rate of change ($r$, blue line) and division rates ($\mu$, red line) for the SAZ and PAZ. Individual points are weekly averaged observations and continuous line is the result of a smoothing temporal filter (Methods). **c**, **d** Averaged time series of the temporal derivative of $\mu$ ($d\mu/dt$, green line) and of the mixed layer depth (MLD) ($dMLD/dt$, magenta line). The blooming phase ($r > 0$) is highlighted by the gray shaded periods. **e**, **f** Bottom maps: Location of float profiles in the SAZ and PAZ.

of the mixed layer dilutes the plankton populations and consequently relaxes phytoplankton mortality rates[8,11,20].

**Subtropical and Seasonal Ice Zones.** The STZ and SIZ represent extreme conditions for the Southern Ocean in terms of their latitudinal location, biogeochemical properties (Fig. S1), and contrasting cycles in the net biomass rate of change (Fig. 3). In the STZ, the annual cycle of $r$ is counterintuitively a near mirror image of the annual cycle in $\mu$ (Fig. 3a), with the blooming phase taking place during months with the lowest mixed layer light levels. What this finding suggests is that variations in the division rate of change ($d\mu/dt$) are not the dominant driver of biomass variability. What we instead find is that net rates of biomass change covary with the rate of change in mixed layer depth ($dMLD/dt$). Thus, the blooming phase ($r > 0$) generally coincides with periods of mixed layer deepening ($dMLD/dt > 0$) and the period of declining biomass corresponds to mixed-layer shoaling

($dMLD/dt < 0$). This pattern suggests a dominant role for the physical impacts of mixing, where deepening of the mixed layer causes a reduction in light-limited phytoplankton division rates, but and even greater decrease in loss (grazing) rates due to the dilution effect discussed above[11,21]. Seasonal changes in mixed layer nutrient availability might also be envisioned as contributing to the unique annual cycle of $r$ for the STZ. However, mixed layer nitrate remains above limiting levels ($>1\,\mu mol\,kg^{-1}$)[22] throughout the year (Fig. S5) and, of our four Southern Ocean zones, the STZ is least associated with iron limiting conditions[23], with mean dissolved iron concentrations remaining above $>0.2$ $nmol\,kg^{-1}$ (Fig. S5). These observations imply that winter enhancements and summer depletion of nutrients likely do not contribute significantly to the unique seasonal cycle in $r$ for the STZ.

Floats used in this study were equipped with ice avoidance software[24], enabling water column sampling beneath ice and thus providing observations throughout the year in the SIZ[25] (Fig. 3b).

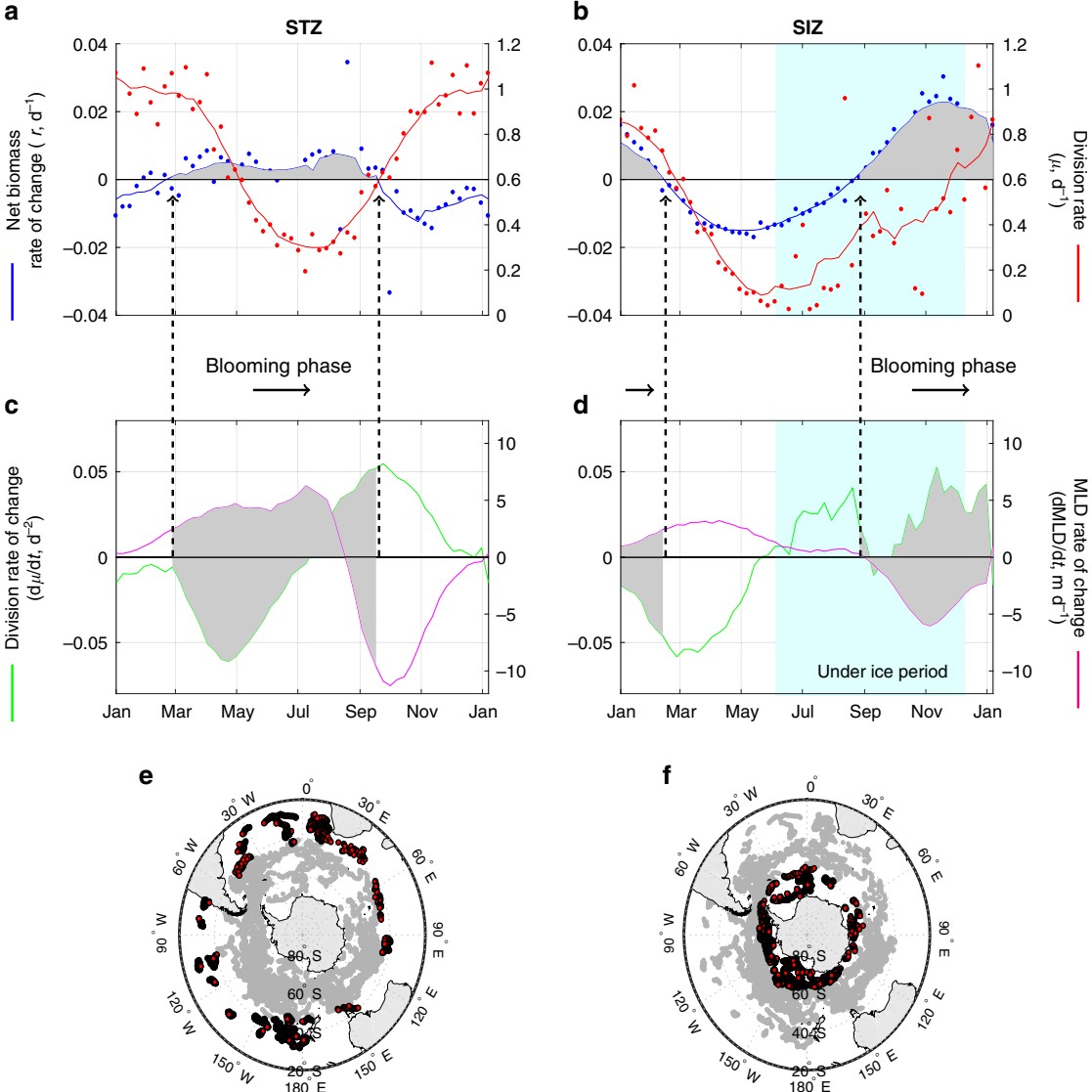

**Fig. 3 Climatological bloom cycles in the Subtropical and Seasonal Ice Zone (STZ and SIZ). a, b** The annual cycle of net phytoplankton biomass rate of change ($r$, blue line) and division rates ($\mu$, red line) for the STZ and SIZ. Individual points are weekly averaged observations and the continuous line is the result of a smoothing temporal filter (Methods). **c, d** Averaged time series of the temporal derivative of $\mu$ ($d\mu/dt$, green line) and of the mixed layer depth (MLD) (dMLD/dt, magenta line). The blooming phase ($r > 0$) is highlighted by the gray shaded periods. The light blue shaded section indicates the period where 50% or more profiles were under ice. **e, f** Bottom maps: Location of float profiles in the STZ and SIZ.

Seasonal cycles in phytoplankton division ($\mu$) and net biomass rates of change ($r$) are similar in the SIZ, with no evident time lag between the two properties. Importantly, under-ice observations in this region documented initiation of the blooming phase prior to ice-out (around September), a phenomenon that has not been accessible through earlier satellite studies of bloom dynamics. Here we define under-ice conditions as times when at least 50% of the float data are from profiles below ice (>30 under-ice profiles per week between June and September during the combined period between 2012 and 2019). Under-ice blooming has been observed at local scales in the Arctic[26] and near Antarctica[27], but our float data set demonstrates that this phenomenon is a common feature of the SIZ. What makes this event particularly remarkable is the low light level at which blooming appears to begin. Specifically, winter mixed layer light levels in the SIZ are estimated here at $<1\,E\,m^{-2}\,d^{-1}$ (Fig. S6) and these values do not include the albedo effect of ice which could reduce these estimates to values close to the compensation level where phytoplankton photosynthesis only supports cellular respiration ($\sim0.04\,E\,m^{-2}$

$d^{-1}$)[28]. Such extreme mixed layer light-limiting conditions only exist in very high polar latitudes such as the SIZ[10] and may explain the tight temporal coupling between $r$ and $\mu$ (i.e., impeding even earlier bloom initiation) observed in this zone exclusively.

**Controls and sensitivity of phytoplankton seasonal bloom in the Southern Ocean.** Light limitation is the dominant factor controlling phytoplankton division in the Southern Ocean, explaining 66% of the variability in division rates ($\mu$) (coefficient of determination, $R^2 = 0.66$, between averaged mixed layer light saturation index and $\mu$, p-value $< 0.05$) (Fig. 4a). However, the magnitude of blooms in the region (i.e., the difference between the mean winter and summer phytoplankton biomass) is correlated with the mean surface dissolved iron concentration (Fig. 4b). This finding is in line with the well-known constraint of iron limitation on biological productivity in the Southern Ocean[23,29,30]. Future changes in surface iron availability could

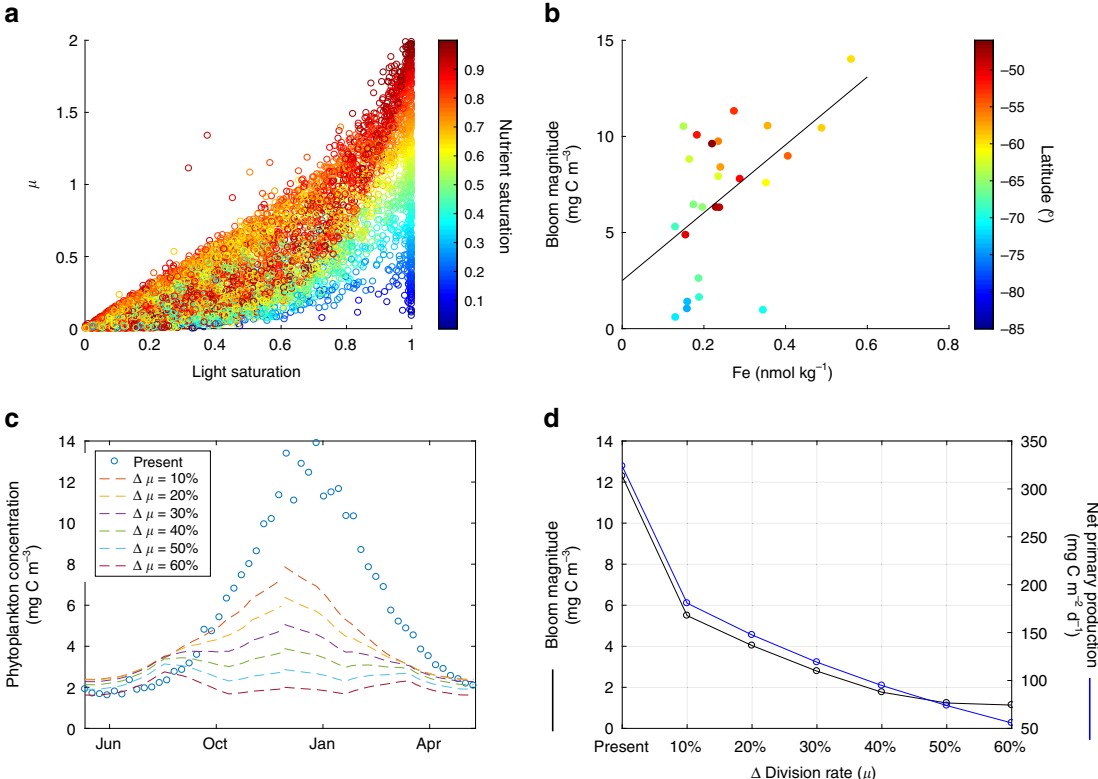

**Fig. 4 Light and iron controls on phytoplankton blooms and future projections of biomass and productivity. a** Relationship between the mean mixed layer phytoplankton division rate ($\mu$) and light and nutrient saturation indices diagnosed by the phytoplankton growth model. **b** Relationship between bloom magnitude and the surface iron concentration in the Southern Ocean. The continuous black line is obtained from a least-squares linear regression model with a coefficient of determination ($R^2$) of 0.26 and a $p$-value = 0.006 ($p$-value corresponds to an $F$-test which evaluates whether the linear model fits significantly better than a degenerate model consisting of only a constant term and slope = 0). **c** Variations in seasonal phytoplankton concentration in the Southern Ocean resulting from a relative decrease (increase) in $\mu$ during summer (winter) with respect to the present division rate. **d** The decrease in mean phytoplankton bloom magnitude (black line and symbols) and annual mean vertically integrated net primary production (NPP, blue line and symbols) in the Southern Ocean as a consequence of relative changes in $\mu$.

thus alter the magnitude of Southern Ocean bloom cycles with respect to present conditions, with implications for marine carbon productivity and export.

Current projections suggest that the Southern Ocean will generally experience an increase in surface ocean stratification in the future[31]. Associated with this intensified stratification will likely be an increase in summer nutrient limitation and a relaxation of winter light limitation[32]. We use our in situ-based estimates of division ($\mu$), loss ($l$), and net biomass rate of change ($r$) to assess the sensitivity of the annual cycle in phytoplankton biomass in the Southern Ocean to changes in division rates by increasing and decreasing $\mu$ over a range from 10 to 60% with respect to current values during winter and summer, respectively. For these simulations, we assumed that loss rates paralleled changes in $\mu$ but with a temporal lag[10] (Methods). The loss rate is recalculated at each sensitivity run based on the resulting seasonal cycle of $\mu$. We find that environmental changes that lead to a decrease in summer division rates tend to reduce bloom magnitude and mean annual productivity despite increased $\mu$ during winter (Figs. 4c, d, and S11). Specifically, bloom magnitude decreases from a mean of 12 mg C m$^{-3}$ for present conditions to 6 mg C m$^{-3}$ for a 10% change in $\mu$ and to ~2 mg C m$^{-3}$ for a 60% change in $\mu$. Similarly, annual mean vertically integrated net primary production (NPP) decreases from 324 mg C m$^{-2}$ d$^{-1}$ for present conditions to 181 and 56 mg C m$^{-2}$ d$^{-1}$ for 10 and 60% changes in $\mu$, respectively.

These results indicate that a relatively small change of 10% in $\mu$ can result in a relatively large (estimated here at ~50%) reduction

in bloom magnitude and NPP. The highly sensitive response of the bloom magnitude and NPP to a reduction in summer $\mu$ results from the fact that division rates decrease during the period of highest (exponential) phytoplankton growth. These estimates are mostly theoretical and are obtained for the average seasonal cycle in phytoplankton biomass in the Southern Ocean, but they respond to the expected general trend in nutrient and light availability if upper ocean stratification is to increase at high latitudes[31,32]. The exact quantitative reduction in bloom magnitude and NPP will, nevertheless, depend on the impact that reduced surface ocean mixing will have on division and loss rates, as well as on the compound net environmental change of the heterogenous Southern Ocean. While the consequences of changes in NPP on oceanic carbon export and sequestration remain to be quantified, our analysis suggests that relatively small changes in phytoplankton division rates in the Southern Ocean could result in 'flatter' seasonal biomass cycles that more closely resemble current lower latitude regions[33].

## Discussion

**Future perspectives on phytoplankton biomass cycles.** Over the last ten years, satellite and restricted in situ studies have shown that phytoplankton biomass often starts increasing in early winter and prior to surface mixed layer shoaling, a finding inconsistent with the classical light-driven interpretation of blooms[10–15]. A new 'Disturbance-Recovery' hypothesis has been proposed that accommodates these findings[11,15], where disturbances such as mixed layer deepening impact predator-prey relationships and

seasonal variations in the division rate of change ($d\mu/dt$) drive changes in phytoplankton concentration over the annual cycle. The development of this hypothesis has largely been based on observations in the northern hemisphere and strongly biased toward satellite, rather than in situ, data. Inferred support for the 'Disturbance-Recovery' hypothesis is also derived from ecological models of the North Atlantic spring bloom[21,34,35]. Here, a large array of biogeochemical floats deployed over the last 7 years has allowed a detailed and in situ evaluation of phytoplankton bloom dynamics in the Southern Ocean. For the region as a whole and for the four subregions investigated, we find that seasonal variations in phytoplankton biomass are well accounted for by the fundamental mechanisms encompassed by the 'Disturbance-Recovery' hypothesis. Nevertheless, it is important to note that direct observations of varying grazing rates as a consequence of changes in mixed layer depth and their net effect on phytoplankton biomass are necessary to confirm this hypothesis. We also find that the relative importance of the balance between disturbances (i.e., dilution of plankton populations by mixed layer deepening) and alterations in the division rate of change ($d\mu/dt$) likely differs across Southern Ocean zones.

Among high latitude regions, the Southern Ocean has major biogeochemical significance, with strong air-sea $CO_2$ fluxes[36,37] and a nutrient supply fueling global marine biological productivity north of 30°S[38]. Understanding the biological engine of the Southern Ocean, and more specifically phytoplankton accumulation and decay cycles (blooms), is therefore key to quantifying biogeochemical fluxes and projecting future changes in marine planktonic ecosystems. Our findings emphasize the important interplay between 'bottom-up' and 'top-down' process and suggest that large changes in carbon biogeochemistry can result from relatively small changes in mixed-layer growth conditions. Continued efforts to better quantify loss rates could provide powerful insights on our understanding of biomass cycles, particularly for discerning the relative role of winter dilution versus nutrient fertilization in regions where the blooming phase is aligned with a deepening of the surface mixed layer. Equipping biogeochemical floats with light sensors would provide both complementary data for comparison with remote sensing information and a unique perspective of the submarine light field experienced by polar phytoplankton under ice. Finally, a refocus in modeling efforts is needed to develop more realistic simulations of both autotroph and heterotroph responses to changes in the physical environment[10,21,39] in order to project with fidelity future changes in phytoplankton biomass cycles and bloom intensity that depart from the current ecological mean state.

## Methods

**Float data.** Quality-controlled float data analyzed in this study were downloaded from the Southern Ocean Carbon and Climate Observations and Modeling (SOCCOM) data portal (https://soccom.princeton.edu/content/data-access). The SOCCOM program is focused on understanding the carbon cycle in the Southern Ocean and determining its influence on climate through the deployment of biogeochemical (BGC)-Argo floats and state-of-the-art climate models. We obtained the 12 March 2019 low-resolution data snapshot (with LIAR-based estimation of carbon chemistry variables, not used) published as a MATLAB data file[40]. The floats are equipped with a CTD (conductivity-temperature-depth), oxygen, nitrate, pH and bio-optical sensors (fluorescence and particulate backscattering at 700 nm ($b_{bp}(700)$)[25]. SOCCOM BGC-Argo floats sample the vertical water column every 5 or 10 days, depending on the preset programming of the float, with most floats sampling every 10 days. The vertical resolution of the measurements taken by the floats varies with depth, with measurements every 5 m in the upper 100 m. The uppermost sampled depth is ~5 or 7 m below surface. Vertical sampling resolution decreases to 10 m below 100 m depth, 20 m below 360 m depth, and 50 m between 400 and 2000 m depth. Vertical profiles are smoothed using a seven point running-median filter. Float data correspond to the period from 06/Mar/2012 to 12/Mar/2019. For the multi-annual time series of the entire Southern Ocean (Fig. 1), we focused on the period from January 2015 onwards, which has sufficient profiles to

permit complete representation of all ocean basins south of 30°S. All other analyses (Figs. 2–4) included the complete data set between 2012 and 2019. All float-based physical and biological variables were initially obtained for each individual float profile (Fig. S2) and subsequently averaged into different zones (Southern Ocean and subregions). The analyses presented were conducted using the scientific programming software MATLAB (version 2017a).

**Estimates of phytoplankton carbon and chlorophyll.** Estimates of particulate organic carbon (POC, mg m$^{-3}$) are obtained based on an empirical relationship established between POC samples taken during float deployment and float measured $b_{bp}(700)$[25,41]:

$$POC = 3.12 \times 10^4 (\pm 2.47 \times 10^3) \times b_{bp}(700) + 3.0 (\pm 6.8) \qquad (2)$$

Phytoplankton carbon ($C_{phyto}$, mg m$^{-3}$) is estimated from an empirical relationship with POC (uncertainties of the empirical relationship are not provided)[42]:

$$C_{phyto} = 0.19 \times POC \pm 8.7 \qquad (3)$$

Chlorophyll concentration (Chl, mg m$^{-3}$) is obtained from float fluorescence measurements corrected for non-photochemical quenching (NPQ) and calibrated against high-performance liquid chromatography (HPLC) measurements based on chlorophyll samples taken during SOCCOM float deployments[25,41]. Float estimates of POC and Chl agree well with satellite ocean color retrievals for the Southern Ocean[41]. For each $C_{phyto}$ profile we subtract the mean estimated concentration between 900 m and 2000 m from the entire vertical profile, in order to make sure that phytoplankton carbon asymptotes towards zero at depth. Resulting negative $C_{phyto}$ concentrations from this subtraction are ≈2% in the entire data set, and <0.001% in the upper 200 m. Negative Chl estimates represent <0.01% of the entire float data set. Negative $C_{phyto}$ and Chl estimates are ultimately removed in order to avoid spurious outputs from the phytoplankton growth model.

**Net biomass rate of change.** The net phytoplankton biomass rate of change ($r$, d$^{-1}$) for each float is computed between observational time-points (profiles) using centered-differences[12]:

$$r\left(t + \frac{\Delta t}{2}\right) \equiv \begin{cases} \frac{1}{\overline{P}}\frac{d\overline{P}}{dt} \approx \frac{2}{\Delta t}\frac{(\overline{P}(t+\Delta t) - \overline{P}(t))}{(\overline{P}(t+\Delta t) + \overline{P}(t))}, & \text{if } \frac{dMLD}{dt} < 0 \\ \frac{1}{\int P}\frac{d\int P}{dt} \approx \frac{2}{\Delta t}\frac{(\int P(t+\Delta t) - \int P(t))}{(\int P(t+\Delta t) + \int P(t))}, & \text{otherwise} \end{cases} \qquad (4)$$

where $t$ is time, $\overline{P}$ is mean $C_{phyto}$ concentration in the mixed layer, and $\int P$ is $C_{phyto}$ integrated from the surface to the bottom of the mixed layer. Equation (4) describes a switching algorithm where $r$ is computed from changes in phytoplankton concentration during periods of mixed layer shoaling and from changes in phytoplankton inventory during periods of mixed layer deepening (or stationary). The aim of Eq. (4) is to remove variations in $r$ not caused by the ecological balance between phytoplankton division rates and losses due to gravitational particle sinking, grazing, or viral infection. Therefore, our net biomass rate of change rate estimates highlight biomass variations driven mainly by ecological processes affecting the accumulation and depletion of phytoplankton. Estimates of $r$ based only on $\overline{P}$ will indicate a decrease in net biomass rate of change during periods of plankton dilution due to mixed layer deepening. Estimates based on $\int P$ alone will indicate a decrease in biomass during periods of mixed layer shoaling due to changes in the vertically integrated water layer. While the overall seasonality of $r$ estimates is based exclusively on $\overline{P}$ or $\int P$ is similar (Fig. S7), differences between $\overline{P}$-based and $\int P$-based estimates of $r$ are observed during mixed-layer shoaling and deepening, consistent with the mechanisms explained above (Fig. S8)[12,43]. Mixed layer depth estimates are obtained using float in situ temperature and salinity profiles[44].

**PAR data.** Estimates of cloud-corrected surface ocean photosynthetically available radiation (PAR, E m$^{-2}$ d$^{-1}$) are obtained form satellite data downloaded from the NASA Ocean Color website (https://oceancolor.gsfc.nasa.gov). Daily global maps of MODIS-Aqua PAR (L3, 4 km) are obtained for the period between the first and last available float profile (06/Mar/2012 and 12/Mar/2019, respectively). Satellite matchups to float profiles are obtained for the same day and the closest pixel to the spatial position of each float profile. If no satellite data is available, NaN is assigned to the corresponding profile PAR matchup. Under ice profiles with unknown locations are also assigned NaN as PAR data matchup. Overall, 77% of float profiles have a valid assigned PAR matchup.

**Dissolved iron data.** Information of dissolved iron (Fe, nmol kg$^{-1}$) is obtained from an updated (June, 2015) version of a global database of dissolved iron observations (available at https://www.bodc.ac.uk/geotraces/data/)[45]. Iron observations are scarce and not gridded. Scattered Fe observations are subsampled by averaging all available observations in the upper 200 m proximate to each float profile within a horizontal radius of 500 km, and taken during the same month as the corresponding float profile.

**Bloom magnitude**. For each available float time series of phytoplankton biomass, mean winter and summer biomass concentrations of phytoplankton carbon (mg C $m^{-3}$) are obtained by averaging the mean mixed layer phytoplankton biomass of all available profiles for the period May–July (winter) and November–January (summer). Bloom magnitude is defined as the difference between the mean winter and summer phytoplankton biomass concentration for each float time series.

**Phytoplankton growth model**. The growth model used here is a modification of the Carbon-based Productivity Model (CbPM)[46]. The CbPM was originally designed to infer vertical profiles of phytoplankton chlorophyll, carbon, division rates and net primary productivity based on satellite estimates of chlorophyll, phytoplankton carbon, and PAR for the surface ocean. We modified the CbPM in order to estimate vertical profiles of phytoplankton division rates ($\mu$, d$^{-1}$) based on float vertical profiles of Chl, $C_{phyto}$, and a satellite-based product of surface PAR. The underwater light field is depth- and spectrally resolved based on satellite surface PAR, float Chl information, and constant spectral fractions from an atmospheric radiative transfer model[47]. The phytoplankton division rate is estimated based on the maximum potential division rate ($\mu_{max} \approx 2$)[48], a nutrient limitation (saturation) term (index) (NSI) constrained by the local Chl:C ratio, and a light limitation (saturation) term (index) (LSI):

$$\mu(z) = \mu_{max} \times \text{NSI}(z) \times \text{LSI}(z) \quad (5)$$

The NSI is inferred from the relative difference between the actual local Chl:C ratio, the Chl:C value when $\mu = 0$, (Chl:$C_{\mu=0} = 3 \times 10^{-4}$)[46], and the theoretical maximum Chl:C achieved under replete nutrient conditions at the local light level (Chl:$C_{max}$)[46,49]:

$$\text{NSI}(z) = \frac{\text{Chl:C}(z) - \text{Chl:C}_{\mu=0}}{\text{Chl:C}_{max} - \text{Chl:C}_{\mu=0}} \quad (6)$$

The nutrient saturation/limitation term is driven by variations of the phytoplankton Chl:C ratio, which is expected to be acclimated to the environmental nutrient and light conditions[50,51]. The model was primarily conceived to diagnose nutrient limitation caused by nitrate depravation[46,49]. Since biological productivity in the Southern Ocean is considered to be iron limited[23,29,30], an important caveat of the growth model used here is that it is not clear how well the Chl:C ratio can represent physiological effects of iron limitation on phytoplankton growth. To a certain degree, iron deprivation should reduce phytoplankton division rates and Chl synthesis, leading to a reduction of Chl:C[52]. Hence, we expect that physiological changes in Chl:C can also serve as an indicator for iron limitation, although iron-stress can result in unique physiological responses that differ from macronutrient stress. Division rates obtained by the CbPM in the Southern Ocean are similar to outputs of $\mu$ obtained from a productivity algorithm parameterized specifically for Southern Ocean waters[16], and compare favorably with estimates of $\mu$ drived from in situ carbon-14 ($^{14}$C) based net primary productivity measurements[17] (Supplementary Information)[19].

The LSI is constrained by the local light level at each depth ($z$)

$$\text{LSI}(z) = 1 - e^{(-5\text{PAR}(z))} \quad (7)$$

**Time series smoothing**. Multi-annual cycles of integrated biomass, mean mixed layer light and depth, as well as $r$ and $\mu$ for the Southern Ocean are produced by sorting in time all available float-based estimates between 2015 and 2019 (Fig. 1). The time series is presented from 2015 onwards since enough data are accumulated at this point to obtain a synoptical view that represents all basins and environmental zones defined within the Southern Ocean. In order to reduce the noise in the temporal signal and obtain a clear seasonal pattern of the blooms, we first smooth the Southern Ocean time series by applying a moving average filter over a 10 days window. Subsequently, we applied a secondary moving filter over 500 consecutive data points to reduce small temporal variability that propagates into the computation of the temporal derivatives. The obtain the mean annual cycle of $r$, $\mu$, d$\mu$/dt, and dMLD/dt in each of the environmental zones (STZ, SAZ, PAZ, and SIZ, Figs. 2 and 3), weekly data for all available years in the float data set (2012–2019) are averaged within each zone, resulting in a weekly resolved annual climatology of all float data (spanning between 2012 and 2019). The annual climatology is subsequently smoothed applying a (single) moving average filter over a 60 days window.

**Environmental zones**. Environmental zones defined in the Southern Ocean[53] are based on a mean 2004–2014 Argo-based climatology of temperature and salinity[54,55] (Fig. S1). The STZ, which roughly covers the oligotrophic oceanic section between 30°S and 40°S, is characterized by reduced surface nutrient concentrations and constrained to the south by the Subtropical Front. The SAZ and PAZ, which cover the circumpolar section of the Southern Ocean approximately constrained between 40°S and 60°S, are characterized by deep mixed layers, high vertical mixing, elevated macronutrient concentrations (i.e., nitrate, phosphate, silicate), and growth-limiting surface iron concentrations[23,29,30]. The SIZ, which represents the seasonally ice-covered zone of the Southern Ocean, extends between Antarctica and ~60°S. Biogeochemical properties in the surface mixed layer sampled by the floats show clear latitudinal gradients across zones[56]: Temperature

decreases from >15 °C in the STZ to ~10 °C in the SAZ and <5 °C towards the SIZ. Mean oxygen in the mixed layer increases from <250 µmol $O_2$ kg$^{-1}$ in the STZ to ~270 µmol $O_2$ kg$^{-1}$ in the SAZ, and >300 µmol $O_2$ kg$^{-1}$ south of the Antarctic polar front. Nitrate also shows a meridional increase from <5 µmol $NO_3$ kg$^{-1}$ in the STZ to >10 µmol $NO_3$ kg$^{-1}$ in the SAZ, and >20 µmol $NO_3$ kg$^{-1}$ south of the polar front, towards the SIZ (Fig. S1).

**Modeling changes in phytoplankton seasonal bloom magnitude**. We assess and quantify theoretical future changes in seasonal phytoplankton bloom magnitude based on present observations of phytoplankton biomass and growth rate parameters inferred from float data:

- The present mean phytoplankton biomass annual cycle in the Southern Ocean is computed by averaging all float-based estimates of mean phytoplankton carbon concentration in the mixed layer on a weekly basis and interpolating them into a daily time series (Fig. 4a). The same procedure is followed to obtain an annual climatology of $r$ and $\mu$.
- Seasonal anomalies in $\mu$ are calculated by subtracting the overall annual mean of $\mu$ from the climatological daily value of $\mu$ ($\mu_{daily} - \mu_{mean}$).
- Relative changes in $\mu$ are computed by decreasing the climatological daily $\mu$ when the seasonal anomaly is positive (larger than the annual mean), and increasing it when the seasonal anomaly is negative (lower than the annual mean) (Fig. S12a). The rationale for this sensitivity exercise is that future increases in ocean stratification should increase nutrient limitation during summer (period of positive anomalies) and relax light limitation during winter (period of negative anomalies). The division rate ($\mu$) is decreased/increased over a range from 10 to 60% with respect to current values during winter and summer, respectively. This exercise does not assess a quantitative relationship between changes in stratification and $\mu$. The goal is to infer changes in seasonal bloom magnitude and NPP as a result of prescribed alterations in $\mu$, given mean climatological values of phytoplankton biomass and $\mu$ derived from float data in the Southern Ocean.
- The net biomass rate of change ($r$) for each scenario is calculated following Eq, (1): $r = \mu - l$. The climatological loss rate ($l$) for each scenario (between 10 and 60 %) is obtained as a 2-days temporally lagged $\mu$. The 2-days lag was determined by reconstructing present net biomass rates of change as $r = \mu - \mu_{xday-lag}$[10], over a range of temporal lags in $\mu_{xday-lag}$ between 1 and 10 days. The best reconstruction of present $r$ was obtained with a temporal lag of 2 days in $\mu$ (Fig. S10).
- Finally, the climatological phytoplankton concentration for each scenario is obtained from a numerical integration of the modeled $r$ using the first value of the current climatological annual cycle as the initial boundary condition (i.e., phytoplankton carbon concentration corresponding to the first day of January).
- Annual cycles of vertically integrated NPP are obtained as the product of climatological division rates ($\mu$) and integrated phytoplankton carbon obtained for each variation of $\mu$ (i.e., between 10 and 60%) (Fig. S12b).

**Reporting summary**. Further information on research design is available in the Nature Research Reporting Summary linked to this article.

## Data availability

BGC-Argo float data are available at the SOCCOM program data portal (https://soccom.princeton.edu/content/data-access). The specific data set used in this work corresponds to the March 12th 2019 low-resolution data snapshot published as a MATLAB data file[40]. Remote sensing PAR data are available at NASA's OceanColor Web (https://oceancolor.gsfc.nasa.gov). Dissolved iron data are available at the GEOTRACES International Data Assembly Centre (https://www.bodc.ac.uk/geotraces/data/).

## Code availability

The phytoplankton growth model (CbPM) code is available at http://sites.science.oregonstate.edu/ocean.productivity/cbpm2.code.php. All analyses were conducted using the scientific programming software MATLAB Version: 9.2.0.538062 (R2017a).

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

## Acknowledgements

This work was supported by the National Aeronautics and Space Administration (NASA) award NNX17AI73G, the Southern Ocean Carbon and Climate Observations and Modeling (SOCCOM) program under the National Science Foundation (NSF) Award PLR-1425989, and the NASA North Atlantic Aerosol and Marine Ecosystem Study (NAAMES).

## Author contributions

L.A.A. developed the study goals and conducted the majority of numerical and data analyses. E.B., M.J.B., and J.L.S. provided guidance and advice in the ecological interpretation of the data and linkage with previous satellite findings. T.K.W. aided in the adaptation of the phytoplankton growth model to float data. All authors contributed to the manuscript text.

## Competing interests

The authors declare no competing interests.
