## [Peer Review File · Nature Communications]

Reviewers' Comments:

Reviewer #1:

Remarks to the Author:

Review

This paper uses the BGC Argo float data from the Southern Ocean to examine the seasonal modulation of phytoplankton biomass and its relationship to phytoplankton division rates. The authors use the lack of concurrence in the phase of these to draw conclusions about phytoplankton loss rates.

The paper shows a nice analysis with valuable findings. But, the authors seem to be set, from the start, to make conclusions about grazing / loss rates, of which they have no information. Also, they keep referring to the annual seasonal cycle of phytoplankton as "blooms" or "bloom events" — which they are not. In many places, they could greatly simplify what they are trying to say, making it much clearer, rather than trying to make a grand claim. I do think the results are worth publishing, but the paper would be much better if it were cleaned up, and the findings were simply stated along with assumptions and uncertainties, without inflated claims.

Title: There is nothing in the paper about ecological drivers. There are no observations of viral losses, grazing, or community. Furthermore, the authors should refrain from calling what they describe in this paper as "blooms" and be clear that they are describing an aggregated signal for the seasonal cycle of phytoplankton biomass and relating it to environmental parameters.

The paper needs to clearly state the assumptions and uncertainties, rather than focus on making claims. For example, there are a lot of caveats to the model for the division rate. It relies on empiricism and many of the parameters used, may not apply to all the BGC floats. This should be made clear. One may question whether the model is appropriate a low light conditions. Under-ice blooms suggest that at high latitudes, growth rates can be significant even in low light conditions at which the cell division rates would be predicted as zero from the model.

Accumulation rate: I found this term very confusing, because the accumulation rate would be in units of phytoplankton biomass gained per day. Instead, what the authors are showing is a "net growth rate" integrated over the water column. I suggest using a different term.

In Equation (1), it would be helpful to add : $dP/dt = r P$.

The authors are then estimating r (integrated over the water column) from $dP/dt / P$ (vertically integrated).

Since $r = \mu - l$, it means that μ should also be the depth-integrated growth rate.

But, μ is the product of μ_{max} , NSI and LSI. This means that the depth integration cannot be done for each term. μ must be evaluated as a function of z and then depth integrated — i.e.

$\mu(z) = \mu_{max} \times LSI(z) \times NSI(z)$ should be depth averaged.

How is μ being calculated in this paper?

The equation $dP/dt = r P$ is actually in the Lagrangian frame. Otherwise, the largest contribution to dP/dt would be due to advection. It could be justified that the Southern Ocean is like a periodic channel, and hence the area-averaged dP/dt does not need to include advection. This means that the data is being used to assess the large-scale average modulation in phytoplankton biomass — and not "blooms". Also, the neglect of advection may not be well justified for the regional calculations. Therefore it is important to be clear about what is being shown — and not claim that the floats are observing blooms.

Phytoplankton division rate μ : it should be made clear that this is a modeled parameter. In other words, r is estimated from measurement, and μ is modeled. Because of the many caveats in modeling μ , the authors should be careful in how they draw conclusions. For example, it would be best to rephrase their claims as contingent on the (light and chl;c based) model for μ .

μ is 100 to 200 times larger than r . The reason, the authors say, is that there is a very fine balance between μ and I . But, in the ocean, one would not expect that such a fine balance can always be maintained, and it is in fact the reason that one might get ephemeral blooms (or events). My concern, here, is that " r " is based on a large-scale average (many floats, over a large region give the biomass, which is differenced in time), The time differencing of 2 large-scale averages, would result in a small r . On the other hand, μ is modeled from point measurements. So I wonder how well the average μ represents the entire region, for comparison with r . Showing the distribution of μ (pdf) in different seasons would be helpful.

Also, distributions (Pdfs) of biomass, Chl:C and MLD would be helpful.

Abstract:

First sentence: Says ... "cannot be fully understood from" ... but needs to say what is missing. E.g. ... because they are also affected by grazing and other losses.

Buoy — this is a misleading term and should be avoided. After explaining that floats are robotic profiling instruments, the authors ought to consistently use the term "float". Information about— how often and how deep/shallow the floats profile, and what they measure— how many floats are averaged, etc. is useful.

The words acceleration and deceleration have been used inappropriately in many places (it is the second derivative w.r.t. time). It would be best to avoid the term or use it carefully.

Line 20: blooming events are not described in this paper (see above), but the annual cycle is analyzed.

Line 20: Declining division rates - may not be clear in the abstract (It should be explained that this is the phytoplankton division rate based on light, c , chl measured by single floats (what depth?). In other words, there are many caveats here and they should be stated.

Line 21: delete "changing"

Line 28: delete "changing phytoplankton" and the sentence will be more clear

Line 38: r is introduced here, as the seasonal changes in biomass. This had me thoroughly confused, as it is not consistent with how it is later calculated. Later r is calculated as $(1/P) dP/dt$.

Line 46: change "drifting buoys" to floats

Lines 50-53 are repeating what is in the abstract.

Line 63: phytoplankton division rates (add " modeled from ... ")

Line 64-67 : I would recommend toning down the claims here - and just stating the findings. E.g.

"clear indication that ... " replace by "suggesting that"

Line 66-67 : see above, comment about μ being 100 times larger

Line 74 "can only be explained" — there might be other explanations in certain situations, like exhausting a nutrient,

Line 76 This paper provides no insights into bloom phenology

Line 77 "rate of change of r " is simpler to understand than "temporal gradient (slope)"

Line 77-78 : Find a less complicated way of explaining DD. E.g. when r has reached a minimum

Line 82-83 : I was not convinced (or could not find an explanation) as to why dP/dt (or do the authors mean r ?) depends on the rate of change of μ . This point was brought up a couple of times and should be justified or explained.

Line 84-87 : there is nothing in this data set to support this, or differentiate between the effect of dilution and zooplankton going into diapause, so it would be better to remove this sentence.

Line 108 : results .

Line 108/ 110 Do the authors really mean "acceleration" and "deceleration" ?

Line 113 typo in this line

Line 123-125 Dilution hypothesis is too speculative

Line 132 : Again, the usage of the terms acceleration and deceleration is inappropriate

Line 138 : Comma missing

Line 139 : do not invoke the dilution hypothesis without more information

Line 151-153 : this is interesting - and it would be good to say more about this

Line 157 : delete "geographically extensive"

Line 195 : delete the word "limited" as it is confusing in the way it appears in the sentence

Line 197 : some of the inconsistency arises due to different interpretations of the term "bloom"

Line 198-210 : Either there needs to be data / analysis to back up this hypothesis, or it should not be introduced. References are also missing. The words acceleration/ deceleration should be avoided.

Line 209 : Avoid acceleration / deceleration

Line 248 : has sufficient

Line 256 : What does this mean? Are the uncertainties in another reference?

Equation (4) What is P (overbear) ? Without knowing, I was not able to see a difference between the two formulations. Presumably, the integration is with respect to z .

Line 322 something is missing

Show the Chl:C pdfs

Line 329 Can should be after ratio

Equation (7) LSI (z) - is a function of z

Fg 4b. Where is the black line?

The section "Future perspectives on phytoplankton bloom cycles" does not have much substance and seems to be a weak part of the paper. I would suggest removing it.

The previous section "Project changes in ..." needs more information or details. The section discusses bloom magnitude, but not seasonality (which is in the section title). It should be explained more clearly, how the bloom magnitude is calculated. The methods section (lines 374-377) is unclear. The first sentence (Lines 167-168) makes a statement that is difficult to verify from Fig. 4a.

Line 370: Is the interpolation linear? Why not use a running mean?

Figure 1. Caption: What is "Average time series"? The x-axis would be better to label with Month (in words) and (4-digit) Year, since 6/15 can be confused with month/date.

Figures 2- 3 : x-axis : Better to span from July to July. Is the month value the 1st or 15th of the month?

y-axis: The units for $dMLD/dt$ would be m per day.

Figure 4 : can be improved. 4b. The black line is not visible. 4d: Which of the curves corresponds to which axis? (colors are not distinguishable).

Caption: relative decrease (increase) in μ . The increase is not clear — Is the increase in the period June-July?

Reviewer #2:

Remarks to the Author:

In this manuscript entitled "Ecological drivers of phytoplankton bloom cycles in the Southern Ocean", the authors combine multi-year observations to determine the key factors governing phytoplankton dynamics in the Southern Ocean. This paper could be an important contribution to the scientific community. However, I have a number of major concerns regarding several points listed below, which will require clarification and probably changes that will need to be taken into account.

Comments:

Figure 1 is for me misleading; I am not sure that it is correct and relevant to try to obtain a single characterization of the annual cycles (biomass and rate...) for the whole Southern Ocean (from 30S to 60S). Especially with the other figures in the manuscript, which reveal important regional differences. In addition, recent literature has shown how the biogeography, productivity and phenology of the Southern Ocean can be complex and diverse (e.g. Thomalla et al. 2011, Ardyna et al. 2017; 2019,

Briggs et al. 2017, Uchida et al. 2019).

Increasing stratification (decreasing MLD) will not have a linear effect on phytoplankton growth across the Southern Ocean. This is supported by two recent studies by Ardyna et al. 2017 and Llorc et al. 2019, which observe a non-monotonic response of phytoplankton biomass to changes in MLD and, more importantly, that a shallowing (or deepening) of the mixed layer is not expected to result in the same biological response everywhere. The scenario proposed here is really too simplistic. Can you try to get a more comprehensive understanding/overview of the potential changes in growth and NPP in different parts of the Southern Ocean?

How can we reconcile what is actually observed with a BGC-Argo float, with blooms that generally last less than a month (Briggs et al. 2018, Uchida et al. 2019, Ardyna et al. 2019) with smoothed annual cycles? Can you really translate what you observe based on these smoothed annual cycles to individual times series of the floats (e.g., a concrete example of an annual cycle of a BGC-Argo float could be a relevant addition to support your conclusion)?

Regarding the smoothed annual cycles, I am concerned about the data processing with three successive smoothing methods (for example the last one covering 60 days, this period is really long especially in the SIZ), will the results remain similar without the smoothing? This is a critical point that should be addressed and included in the SI.

Furthermore, I do not understand why the authors use only 4 years of data for the BGC-Argo floats. The use of all years and all available data will definitively improve the characterization of the different provinces and will even give some indication of intra-regional variability. I also noticed that the "old" Geotraces 2015 dataset was used, please update it with the more recent Geotraces 2017 dataset (<https://www.bodc.ac.uk/geotraces/data/idp2017/>), to improve Fig S2 and Fig 4b.

References:

- Ardyna, M., H. Claustre, J.-B. Sallée, F. d'Ovidio, B. Gentili, G. van Dijken, F. D'Ortenzio, and K. Arrigo. 2017. Delineating environmental control of phytoplankton biomass and phenology in the Southern Ocean. *Geophys. Res. Lett.* 44:5016–5024.
- Ardyna, M., L. Lacour, S. Sergi, F. d'Ovidio, J.-B. Sallée, M. Rembauville, S. Blain, A. Tagliabue, R. Schlitzer, C. Jeandel, K. R. Arrigo, and H. Claustre. 2019. Hydrothermal vents trigger massive phytoplankton blooms in the Southern Ocean. *Nat. Commun.* 10:2451.
- Briggs, E. M., T. R. Martz, L. D. Talley, M. R. Mazloff, and K. S. Johnson. 2017. Physical and Biological Drivers of Biogeochemical Tracers Within the Seasonal Sea Ice Zone of the Southern Ocean From Profiling Floats. *J. Geophys. Res. Oceans.*
- Llorc, J., M. Lévy, J. B. Sallée, and A. Tagliabue. 2019. Non-monotonic response of primary production and export to changes in mixed-layer depth in the Southern Ocean. *Geophys. Res. Lett.*
- Thomalla, S. J., N. Fauchereau, S. Swart, and P. M. S. Monteiro. 2011. Regional scale characteristics of the seasonal cycle of chlorophyll in the Southern Ocean. *Biogeosciences* 8:2849-2866.
- Uchida, T., D. Balwada, R. Abernathy, C. J. Prend, E. Boss, and S. T. Gille. 2019. Southern Ocean Phytoplankton Blooms Observed by Biogeochemical Floats. *Journal of Geophysical Research: Oceans* 124:7328-7343.

Dear editors and reviewers,

Please find below our detailed responses to the comments made by the reviewers on the manuscript now titled “Seasonal modulation of phytoplankton biomass in the Southern Ocean”, submitted to *Nature Communications*. We would like to take the opportunity to thank the reviewers for the effort they put into reviewing our manuscript, as well as for the valuable feedback leading to the improvement of the manuscript. Please also find attached a document with highlighted changes between the previous and revised versions of the manuscript.

Reviewer (R) 1

General comments

1) R1: “This paper uses the BGC Argo float data from the Southern Ocean to examine the seasonal modulation of phytoplankton biomass and its relationship to phytoplankton division rates. The authors use the lack of concurrence in the phase of these to draw conclusions about phytoplankton loss rates.

The paper shows a nice analysis with valuable findings. But, the authors seem to be set, from the start, to make conclusions about grazing / loss rates, of which they have no information. Also, they keep referring to the annual seasonal cycle of phytoplankton as blooms or bloom events which they are not. In many places, they could greatly simplify what they are trying to say, making it much clearer, rather than trying to make a grand claim. I do think the results are worth publishing, but the paper would be much better if it were cleaned up, and the findings were simply stated along with assumptions and uncertainties, without inflated claims.

Title: There is nothing in the paper about ecological drivers. There are no observations of viral losses, grazing, or community. Furthermore, the authors should refrain from calling what they describe in this paper as blooms and be clear that they are describing an aggregated signal for the seasonal cycle of phytoplankton biomass and relating it to environmental parameters.

The paper needs to clearly state the assumptions and uncertainties, rather than focus on making claims. For example, there are a lot of caveats to the model for the division rate. It relies on empiricism and many of the parameters used, may not apply to all the BGC floats. This should be made clear. One may question whether the model is appropriate a low light conditions. Under-ice blooms suggest that at high latitudes,

growth rates can be significant even in low light conditions at which the cell division rates would be predicted as zero from the model.”

Reply: In the present analysis, we evaluate seasonal cycles in phytoplankton biomass over large regions. This approach effectively captures large-scale regionally-averaged modulations in phytoplankton biomass and, herein, we refer to periods of the year when maximal biomass occurs as the “bloom”, consistent with earlier treatments in peer-reviewed literature (Longhurst, 1993; Siegel et al., 2002; Boss and Behrenfeld, 2010; Behrenfeld and Boss, 2018; Behrenfeld et al., 2017), as well as textbooks (Miller, 2004; Mann and Lazier, 2006). In practical terms, we refer to the “blooming phase” as periods of time when the net phytoplankton biomass rate of change (r) > 0 (previously termed as “accumulation rate”, see Reply 2 below). Our calculations of net biomass rate of change (r) and division rates (μ) are conducted individually for each float time series (Figure 1 this review and Figure S2 in the revised manuscript (ms)), and averaged over the Southern Ocean domain (south of 30°S, and subregions) to present a synoptic view of the seasonal modulation in phytoplankton biomass in the Southern Ocean. However, the reviewer makes a good point and we note that the amplitude and duration of our spatially-integrated blooms are smaller and longer, respectively, than is often observed in the field at local scales. The same is true for the spatially-integrated environmental variables used here to understand primary drivers of the phytoplankton biomass annual cycles. We have retitled our manuscript as “Seasonal modulation of phytoplankton biomass in the Southern Ocean”, to better represent the large-scale synoptic scope of our study. We now clarify in the beginning of the manuscript that “*Phytoplankton blooms are here defined as large-scale regionally-averaged positive periods of net rate of change of phytoplankton biomass ($r > 0$), , which differs from short time-scale blooming events that are often observed at small spatial scales in the field*” (line 62–64 of the revised ms). Other concerns related to the computation and interpretation of r and μ are further addressed below.

2) R1: “Accumulation rate: I found this term very confusing, because the accumulation rate would be in units of phytoplankton biomass gained per day. Instead, what the authors are showing is a net growth rate integrated over the water column. I suggest using a different term.”

Reply: We modified the manuscript to introduce “ r ” as the “biomass-specific net rate of change”. To avoid further confusion, we use the terms “division rate” for μ and “net rate of change of phytoplankton biomass” for r , which is consistent with the terminology in recent phytoplankton bloom studies (Behrenfeld et al., 2017; Mignot et al., 2018; Behrenfeld and Boss, 2018).

3) R1: “In Equation (1), it would be helpful to add : $dP/dt = r P$. The authors are then estimating r (integrated over the water column) from $dP/dt / P$ (vertically integrated). Since $r = \mu - l$, it means that μ should also be the depth-integrated growth rate. But, μ is the product of μ_{max} , NSI and LSI. This means that the depth integration cannot be done for each term. μ must be evaluated as a function of z and then depth integrated i.e. $\mu(z) = \mu_{\text{max}} \times \text{LSI}(z) \times \text{NSI}(z)$ should be depth averaged. How is μ being calculated in this paper?”

The equation $dP/dt = r P$ is actually in the Lagrangian frame. Otherwise, the largest contribution to dP/dt

would be due to advection. It could be justified that the Southern Ocean is like a periodic channel, and hence the area-averaged dP/dt does not need to include advection. This means that the data is being used to assess the large-scale average modulation in phytoplankton biomass and not blooms. Also, the neglect of advection may not be well justified for the regional calculations. Therefore it is important to be clear about what is being shown and not claim that the floats are observing blooms.

Phytoplankton division rate μ : it should be made clear that this is a modeled parameter. In other words, r is estimated from measurement, and μ is modeled. Because of the many caveats in modeling μ , the authors should be careful in how they draw conclusions. For example, it would be best to rephrase their claims as contingent on the (light and chl:c based) model for μ .”

Reply: We have modified Equation 1 to show that r is the biomass-specific net rate of change of phytoplankton:

$$r = \frac{1}{P} \frac{dP}{dt} = \mu - l$$

The phytoplankton growth model is modified to estimate vertically-resolved μ , as stated in section “Methods”, subsection “Phytoplankton growth model”: “*We modified the CbPM in order to estimate vertical profiles of phytoplankton division rates (μ , d^{-1}) based on float vertical profiles of Chl, C_{phyto} , and a satellite-based product of surface PAR.*” (line 334 – 336). We have modified Equations 5, 6 and 7, to highlight the depth dependence of μ , NSI, and LSI:

$$\mu(z) = \mu_{\max} \times \text{NSI}(z) \times \text{LSI}(z)$$

$$\text{NSI}(z) = \frac{\text{Chl:C}(z) - \text{Chl:C}_{\mu=0}}{\text{Chl:C}_{\max} - \text{Chl:C}_{\mu=0}}$$

$$\text{LSI}(z) = 1 - e^{(-5\text{PAR}(z))}$$

Division rates (μ) are averaged for the mixed layer in order to be compared in magnitude and timing with the net rate of change of phytoplankton biomass (r). In line 71 of the revised ms, we clarify that presented μ is obtained from “*mixed layer-averaged modeled phytoplankton division rates*”. The caveats and considerations of the growth model (CbPM) are discussed in subsection “Phytoplankton growth model”, lines (345–354): “*The nutrient saturation/limitation term is driven by variations of the phytoplankton Chl:C ratio, which is expected to be acclimated to the environmental nutrient and light conditions (Geider, 1987; Arteaga et al., 2014). The model was primarily conceived to diagnose nutrient limitation caused by nitrate deprivation (Behrenfeld et al., 2005; Westberry et al., 2008). Since biological productivity in the Southern Ocean is considered to be iron limited (Martin et al., 1990; Boyd et al., 2007; Moore et al., 2013), an important caveat of the growth model used here is that it is not clear how well the Chl:C ratio can represent physiological effects of iron limitation on phytoplankton growth. To a certain degree, iron deprivation should reduce phytoplankton division rates and Chl synthesis, leading to a reduction of Chl:C (Geider and LaRoche, 1994). Hence, we expect that physiological changes in Chl:C can also serve as an indicator for iron limitation, although iron-stress can result in unique physiological responses that differ from macronutrient stress (Behrenfeld and*

Milligan, 2013).”

Please see Reply 1 regarding the interpretation of blooms in the context of this study. The general seasonality in phytoplankton division rates (μ) and net biomass rate of change (r) observed in the aggregated Southern Ocean data (Figure 1 of the revised ms) can also be observed in individual float time series with multi-annual observations (newly added Figure S3 in the revised ms). The temporal misalignment between μ and r is observed in individual float time series, where the seasonal cycle of μ and r becomes more aligned for floats further south, closer to the Seasonal Ice Zone (SIZ), consistent with what is observed in the weekly-averaged seasonal climatologies (Figure 2 and 3 of the revised ms).

4) R1: “mu is 100 to 200 times larger than r . The reason, the authors say, is that there is a very fine balance between mu and l . But, in the ocean, one would not expect that such a fine balance can always be maintained, and it is in fact the reason that one might get ephemeral blooms (or events). My concern, here, is that r is based on a large-scale average (many floats, over a large region give the biomass, which is differenced in time), The time differencing of 2 large-scale averages, would result in a small r. On the other hand, mu is modeled from point measurements. So I wonder how well the average mu represents the entire region, for comparison with r. Showing the distribution of mu (pdf) in different seasons would be helpful. Also, distributions (Pdfs) of biomass, Chl:C and MLD would be helpful.”

Reply: As discussed above (Reply 1), the net rate of change of phytoplankton biomass (r) is computed for each individual float time series and averaged over the Southern Ocean and subregions. The two orders of magnitude difference between μ and r is observed in individual floats time series of the Southern Ocean (new Figure S3) as well as in the North Atlantic (e.g., Figure 3 of Boss and Behrenfeld, 2010), and thus, it is not a result of large scale averaging. Nevertheless, it should be noted that the amplitude of the large scale spatially-averaged bloom tends to be smaller than local deviations observed in the field, as discussed above. The seasonal normalized frequency of μ , r , and other float-based variables are shown in Figure 1 of this reply document. This figure has also been added to the Supplementary Information of the revised ms (Figure S2).

Specific comments from R1

5) R1: “Abstract: First sentence: Says cannot be fully understood from but needs to say what is missing. E.g. because they are also affected by grazing and other losses.”

Reply: We have modified this sentence to (lines 15–17 of revised ms):“... *cannot be fully understood from environmental properties controlling phytoplankton division rates (e.g., nutrients and light), as they omit the role of ecological and environmental loss processes (e.g., grazing, viruses, sinking).*”

6) R1: “Buoy this is a misleading term and should be avoided. After explaining that floats are robotic profiling instruments, the authors ought to consistently use the term float. Information about how often and how deep/shallow the floats profile, and what they measure how many floats are averaged, etc. is useful”

Figure 1: Normalized frequency distribution of float-based phytoplankton variables μ , r , biomass (phytoplankton carbon), Chl:C ratio, and mixed layer depth (MLD), for each season (winter (blue), spring (red), summer (green), and autumn (black)) in the entire Southern Ocean float data set.

Reply: We have removed “buoy(s)” and kept “float(s)” throughout the revised ms. Detailed information on the profiling features of the floats is on the “Methods” section, subsection “Float data”: “*Quality-controlled float data analyzed in this study were downloaded from the Southern Ocean Carbon and Climate Observations and Modeling (SOCCOM) data portal (<http://soccompu.princeton.edu/www/index.html>). The SOCCOM program is focused on understanding the carbon cycle in the Southern Ocean and determining its influence on climate through the deployment of biogeochemical (BGC)-Argo floats and state-of-the-art climate models. We obtained the March 12th 2019 low resolution data snapshot (with LIAR-based estimation of carbon chemistry variables, not used) published as a MATLAB data file (Johnson et al., 2019). The floats are equipped with a CTD (conductivity-temperature-depth), oxygen, nitrate, pH and bio-optical sensors (fluorescence and particulate backscattering at 700 nm ($b_{bp}(700)$)) (Johnson et al., 2017). SOCCOM BGC-Argo floats sample the vertical water column every 10 or 5 days, depending on the preset programming of the float, with most floats sampling every 10 days. The vertical resolution of the measurements taken by the floats varies with depth, with measurements every 5 m in the upper 100 m. The uppermost sampled depth is ~ 5 or 7 m below surface. Vertical sampling resolution decreases to 10 m below 100 m depth, 20*

m below 360 m depth, and 50 m between 400 and 2000 m depth. Vertical profiles are smoothed using a seven point running-median filter. Float data correspond to the period from 06/Mar/2012 to 12/Mar/2019. For the multi-annual time series of the entire Southern Ocean (Figure 1), we focused on the period from January 2015 onwards, which has sufficient profiles to permit complete representation of all ocean basins south of 30° S. All other analyses (Figures 2–4) included the complete data set between 2012–2019. All float-based physical and biological variables were initially obtained for each individual float profile (Figure S2) and subsequently averaged into different zones (Southern Ocean and subregions). The analyses presented were conducted using the scientific programming software MATLAB (version 2017a)."

7) R1: "The words acceleration and deceleration have been used inappropriately in many places (it is the second derivative w.r.t. time). It would be best to avoid the term or use it carefully."

Reply: We replace "Decline deceleration" for "*r* Minima (*r*M) " in Figure 1 of the revised ms. We also clarify that the terms acceleration/deceleration refer to times when $d\mu/dt > 0$ or $d\mu/dt < 0$, respectively, where $d\mu/dt$ has units of d^{-2} (lines 117 and 120 of the revised ms). We avoid the use of these terms (acceleration/deceleration) in other parts of the revised ms.

8) R1: "Line 20: blooming events are not described in this paper (see above), but the annual cycle is analyzed."

Reply: As discussed above (Reply 1), we clarify the definition of blooms in the context of large scale seasonal features of positive net phytoplankton biomass rate of change (line 62–64 revised ms).

9) R1: "Line 20: Declining division rates - may not be clear in the abstract (It should be explained that this is the phytoplankton division rate based on light, c, chl measured by single floats (what depth?). In other words, there are many caveats here and they should be stated."

Reply: We have modified this sentence to "modeled division rates" (line 21). The discussion of model caveats is addressed above (Reply 3).

10) R1: "Line 21: delete changing"

Reply: We removed "changing" in the revised ms.

11) R1: "Line 28: delete changing phytoplankton and the sentence will be more clear"

Reply: We removed "changing phytoplankton" in the revised ms.

12) R1: "Line 38: *r* is introduced here, as the seasonal changes in biomass. This had me thoroughly confused, as it is not consistent with how it is later calculated. Later *r* is calculated as $(1/P) dP/dt$."

Reply: As discussed above (Reply 2), r is now introduced as the biomass-specific net rate of change, calculated as $r = \frac{1}{P} \frac{dP}{dt} = \mu - l$ (line 40 and Equation 1 of revised ms).

13) R1: “Line 46: change drifting buoys to floats.”

Reply: The term “buoy(s)” has been replaced for “float(s)” throughout the manuscript.

14) R1: “Lines 50-53 are repeating what is in the abstract.”

Reply: We believe that these sentences are important to introduce the last section of results of the study.

15) R1: “Line 63: phytoplankton division rates (add modeled from)”

Reply: As discussed above (Reply 3), here we have introduced: “*mixed layer-averaged modeled phytoplankton division rates*” (line 71 of revised ms).

16) R1: “Line 64-67 : I would recommend toning down the claims here - and just stating the findings. E.g. clear indication that replace by suggesting that”

Reply: We have replaced “clear indication” by “suggesting that” (line 72 of revised ms).

17) R1: “Line 66-67: see above, comment about mu being 100 times larger”

Reply: Please see Reply 4.

18) R1: “Line 74 can only be explained there might be other explanations in certain situations, like exhausting a nutrient,”

Reply: Nutrient exhaustion could certainly drive variations in μ . However, temporal changes in r that are not reflected in μ (i.e., temporal misalignment) need to be driven by changes in the net balance between μ and l .

19) R1: “Line 76 This paper provides no insights into bloom phenology”

Reply: We replaced “phenology” for “seasonal cycles of biomass” in the revised ms.

20) R1: “Line 77 rate of change of r is simpler to understand that temporal gradient (slope)”

Reply: We replaced “temporal gradient” for “rate of change” in the revised ms.

21) R1: “Line 77-78 : Find a less complicated way of explaining DD. E.g. when r has reached a minimum.”

Reply: DD has been replaced by “ r_M ” in the revised ms. The explanation of r_M has been changed to: “*The moment when r stops decreasing (but is still < 0) marks the time in autumn when the net loss of biomass is highest (minimum r , r_M) (Figure 1).*” (lines 85–87 of revised ms).

22) R1: “Line 82-83: I was not convinced (or could not find an explanation) as to why dP/dt (or do the authors mean r ?) depends on the rate of change of μ . This point was brought up a couple of times and should be justified or explained.”

Reply: This statement refers to r , and it is based on the empirical observation that seasonal variations in r correlate with seasonal variations in $d\mu/dt$ and not with μ (observed in Figure 2 of the revised ms, as well as in Behrenfeld et al., 2017). We clarify this in the revised manuscript: “*These findings imply that the net phytoplankton biomass rate of change (r) is not dependent on the absolute value of μ , but rather on the rate of change in μ .*” (lines 88–90 of revised ms). This observations is also discussed regarding the seasonal cycle of the PAZ and SAZ (Figure 2): “*In contrast, the annual cycle in r is temporally aligned with that of the division rate of change ($d\mu/dt$, i.e., the temporal derivative of μ).*” (lines 114–115 of the revised ms). As discussed in lines 90–92, we believe that “*Such a relationship will exist when division and loss rates are tightly coupled, but a temporal lag exists in the response time for the loss processes (Behrenfeld, 2014; Behrenfeld et al., 2017; Behrenfeld and Boss, 2018).*”

23) R1: “Line 84-87 : there is nothing in this data set to support this, or differentiate between the effect of dilution and zooplankton going into diapause, so it would be better to remove this sentence.”

Reply: We have removed this sentence from the revised ms.

24) R1: “results .”

Reply: We deleted the extra space.

25) R1: “Line 108/ 110 Do the authors really mean acceleration and deceleration ?”

Reply: Yes, please see Reply 7.

26) R1: “typo in this line”

Reply: We fixed the typo in this section of the revised ms.

27) R1: “Line 123-125 Dilution hypothesis is too speculative”

Reply: We understand that the link to dilution as a mechanistic driver cannot be proven with our float data alone. However, it should be noted that the ‘Disturbance-Recovery’ hypothesis does not exclude other

“disturbances” that could alter the balance between grazing and phytoplankton growth. In this case, the “dilution effect” is consistent with the timing of events in μ , r , and seasonal changes in MLD, and has been proposed as a potential mechanism of “disturbance” in upper ocean planktonic ecosystems multiple times in the literature (Behrenfeld, 2010; Banse, 1992, 1994; Behrenfeld, 2014; Behrenfeld et al., 2017; Behrenfeld and Boss, 2018; Yang et al., 2020). Please see also see Reply 31 for more discussion regarding this point.

28) R1: “Line 132 : Again, the usage of the terms acceleration and deceleration is inappropriate”

Reply: We have replaced these terms for the following sentence: “ *What this finding suggests is that variations in the division rate of change ($d\mu/dt$) are not the dominant driver of biomass variability.*” (lines 141–143 of revised ms).

29) R1: “Line 138 : Comma missing”

Reply: Comma added in this section of the revised ms.

30) R1: “Line 139 : do not invoke the dilution hypothesis without more information”

Reply: Please see Reply 24 and Reply 31.

31) R1: “Line 151-153 : this is interesting - and it would be good to say more about this”

Reply: In the revised ms we highlight the occurrence of the under-ice bloom despite very low inferred light levels. Further analysis is in progress to assess the biogeochemical relevance of this under-ice bloom/seasonal biomass increase.

32) R1: “Line 157 : delete geographically extensive”

Reply: We deleted “geographically extensive” in this section of the ms.

33) R1: “Line 195 : delete the word limited as it is confusing in the way it appears in the sentence”

Reply: We replaced “limited” for “restricted” in this section of the ms.

34a) R1: “Line 197 : some of the inconsistency arises due to different interpretations of the term bloom”

34b) R1: “Line 198-210 : Either there needs to be data / analysis to back up this hypothesis, or it should not be introduced. References are also missing. The words acceleration/ deceleration should be avoided.”

Reply: We have modified this section to provide further observational and model-based support to the ‘Disturbance-Recovery’ hypothesis. Furthermore, we clarify that direct observational evidence of seasonal

changes in grazing rates is still needed to fully validate this hypothesis (lines 205–225):

“Over the last ten years, satellite and restricted in situ studies have shown that phytoplankton biomass often starts increasing in early winter and prior to surface mixed layer shoaling, a finding inconsistent with the classical light-driven interpretation of blooms (Boss and Behrenfeld, 2010; Behrenfeld, 2010; Mignot et al., 2018; Behrenfeld and Boss, 2018; Westberry et al., 2016; Behrenfeld et al., 2017). A new ‘Disturbance-Recovery’ hypothesis has been proposed that accommodates these findings, where disturbances such as mixed layer deepening impact predator-prey relationships and seasonal variations in the division rate of change ($d\mu/dt$) drive changes in phytoplankton concentration over the annual cycle. The development of this hypothesis has largely been based on observations in the northern hemisphere and strongly biased toward satellite, rather than in situ, data. Inferred support for the ‘Disturbance-Recovery’ hypothesis is also derived from ecological models of the North Atlantic spring bloom (Behrenfeld et al., 2013; Smith et al., 2015; Yang et al., 2020). Here, a large array of biogeochemical floats deployed over the last 7 years has allowed a detailed and in situ evaluation of phytoplankton bloom dynamics in the Southern Ocean. For the region as a whole and for the four subregions investigated, we find that seasonal variations in phytoplankton biomass are well accounted for by the fundamental mechanisms encompassed by the ‘Disturbance-Recovery’ hypothesis. Nevertheless, it is important to note that direct observations of varying grazing rates as a consequence of changes in mixed layer depth and their net effect on phytoplankton biomass are necessary to confirm this hypothesis. We also find that the relative importance of the balance between disturbances (i.e., dilution of plankton populations by mixed layer deepening) and alterations in the division rate of change ($d\mu/dt$) likely differs across Southern Ocean zones.”

35) R1: “Line 209 : Avoid acceleration / deceleration”

Reply: Acceleration/ deceleration have been replaced by “...alterations in the division rate of change ($d\mu/dt$)...” (line 224 of revised ms).

36) R1: “Line 248 : has sufficient”

Reply: “a” was removed in “has a sufficient” in this section of the ms.

37) R1: “Line 256 : What does this mean? Are the uncertainties in another reference?”

Reply: Uncertainties for the C_{phyto} regression in Graff et al. (2015) are not provided.

38) R1: “Equation (4) What is P (overbear) ? Without knowing, I was not able to see a difference between the two formulations. Presumably, the integration is with respect to z.”

Reply: The meaning of these symbols is indicated right below Equation 4 of the revised ms (line 289–290): “... \bar{P} is mean C_{phyto} concentration in the mixed layer, and $\int P$ is C_{phyto} integrated from surface to the bottom of the mixed layer.”

39) R1: “Line 322 something is missing”

Reply: We are not sure what the reviewer was referring to here.

40) R1: “Show the Chl:C pdfs”

Reply: Please see Figure 1 of this review document (also added as Figure S2 in the revised ms).

41) R1: “Line 329 Can should be after ratio”

Reply: “can” has been moved after “Chl:C ratio”.

42) R1: “Equation (7) LSI (z) - is a function of z”

Reply: Correct. We have incorporated this changes in equations 5, 6, and 7 (see also Reply 3).

43) R1: “Fg 4b. Where is the black line?”

Reply: Please see Reply 48 below.

44) R1: “The section Future perspectives on phytoplankton bloom cycles does not have much substance and seems to be a weak part of the paper. I would suggest removing it. The previous section Project changes in ... needs more information or details. The section discusses bloom magnitude, but not seasonality (which is in the section title). It should be explained more clearly, how the bloom magnitude is calculated. The methods section (lines 374-377) is unclear. The first sentence (Lines 167-168) makes a statement that is difficult to verify from Fig. 4a.”

Reply: section “Future perspectives on phytoplankton biomass cycles”: We have modified this section to provide greater context for the merits and caveats of considering the ‘Disturbance-Recovery’ hypothesis to explain observed patterns of phytoplankton seasonal cycles (i.e., large scale blooms) (see Reply 31). We believe this section serves as a conclusion for the study and lists several important aspects to consider towards the future assessment of planktonic ecosystems.

We have renamed section “Projected changes in phytoplankton bloom seasonality and magnitude” to “Projected changes in phytoplankton seasonal bloom magnitude”.

Section “Methods”, subsection “Bloom magnitude”, is modified in the revised ms to improve clarity of the bloom magnitude calculation: “*For each available float time series of phytoplankton biomass, mean winter and summer biomass concentrations of phytoplankton carbon (mg C m^{-3}) are obtained by averaging the mean mixed layer phytoplankton biomass of all available profiles for the period May–July (winter) and November*”

– January (summer). Bloom magnitude is defined as the difference between the mean winter and summer phytoplankton biomass concentration for each float time series.”

We have modified and retitled subsection “Modeling changes in phytoplankton seasonal bloom magnitude” to improve the description of the assessment of future changes in bloom magnitude and net primary production:

“Modeling changes in phytoplankton seasonal bloom magnitude

We assess and quantify theoretical future changes in seasonal phytoplankton bloom magnitude based on present observations of phytoplankton biomass and growth rate parameters inferred from float data:

1. *The present mean phytoplankton biomass annual cycle in the Southern Ocean is computed by averaging all float-based estimates of mean phytoplankton carbon concentration in the mixed layer on a weekly basis and interpolating them into a daily time series (Figure 8a). The same procedure is followed to obtain an annual climatology of r and μ .*
2. *Seasonal anomalies in μ are calculated by subtracting the climatological daily value of μ from the overall annual mean of μ .*
3. *Relative changes in μ are computed by decreasing the climatological daily μ when the seasonal anomaly is positive (larger than the annual mean), and increasing it when the seasonal anomaly is negative (lower than the annual mean) (Figure S9a). The rationale for this sensitivity exercise is that future increases in ocean stratification should increase nutrient limitation during summer (period of positive anomalies) and relax light limitation during winter (period of negative anomalies). The division rate (μ) is decreased/increased over a range from 10% to 60% with respect to current values during winter and summer, respectively. This exercise does not assess a quantitative relationship between changes in stratification and μ . The goal is to infer changes in seasonal bloom magnitude and net primary production as a results of prescribed alterations in μ , given mean climatological values of phytoplankton biomass and μ derived from float data in the Southern Ocean.*
4. *The net biomass rate of change (r) for each scenario is calculated following Equation 1: $r = \mu - l$. The climatological loss rate (l) for each scenario (between 10 and 60 %) is obtained as a 2-days temporally lagged μ . The 2-days lag was determined by reconstructing present net biomass rates of change as $r = \mu - \mu_{\text{day-lag}}$ (Behrenfeld and Boss, 2018), over a range of temporal lags in $\mu_{\text{day-lag}}$ between 1 and 10 days. The best reconstruction of present r was obtained with a temporal lag of 2 days in μ (Figure S8).*
5. *Finally, the climatological phytoplankton concentration for each scenario is obtained from a numerical integration of the modeled r using the first value of the current climatological annual cycle as the initial boundary condition (i.e., phytoplankton carbon concentration corresponding to the first day of January).*

6. *Annual cycles of vertically integrated net primary production are obtained as the product of climatological division rates (μ) and integrated phytoplankton carbon obtained for each variation of μ (i.e., between 10 and 60 %) (Figure S9b).*”

The statement in line 178 of the revised ms is based on the coefficient of determination ($R^2 = 0.66$) between the averaged mixed layer light saturation and μ from inferred from all profiles for the Southern Ocean. This clarification has been added in the revised ms.

45) R1: “Line 370: Is the interpolation linear? Why not use a running mean?”

Reply: To derive an annual climatology, available Southern Ocean float data are sufficient to compute robust mean weekly estimates and use a linear interpolation to the daily temporal scale.

46) R1: “Figure 1. Caption: What is Average time series? The x-axis would be better to label with Month (in words) and (4-digit) Year , since 6/15 can be confused with month/date.”

Reply: Figure 1: We clarify the average time series in the caption of the revised ms: “*Averaged time series (a – c) are based on individual float-based observations for the Southern Ocean. See Methods for details on the smoothing of time series.*”. We have changed the x-axis label to month (in words)/year (4-digit).

47) R1: “Figures 2- 3 : x-axis : Better to span from July to July. Is the month value the 1st or 15th of the month? y-axis: The units for dMLD/dt would be m per day.”

Reply: Figure 2 and 3: We have experimented before with spanning these panels from June to July, and this complicates the interpretation of the STZ (see Figure 2 of this review as example). Hence, in order to keep consistency, we have kept all panels from Jan to Dec. The grid is located at the 1st of the month. We have fixed the units of dMLD/dt to m d^{-1} .

48) R1: “Figure 4 : can be improved. 4b. The black line is not visible. 4d: Which of the curves corresponds to which axis? (colors are not distinguishable). Caption: relative decrease (increase) in μ . The increase is not clear Is the increase in the period June-July?”

Reply: Figure 4b: Black regression line has been made thicker. Figure 4d: Line colors next to axis labels have been made thicker: Bloom magnitude (black, left), and NPP (blue, right). The decrease (increase) in μ is observed in Figure S9(a). Decreases in μ occur between September and February, while increases occur between March and August.

Figure 2: Subpanel of STZ spanning from June to April

Reviewer (R) 2

General comments

1) R2: “In this manuscript entitled ”Ecological drivers of phytoplankton bloom cycles in the Southern Ocean”, the authors combine multi-year observations to determine the key factors governing phytoplankton dynamics in the Southern Ocean. This paper could be an important contribution to the scientific community. However, I have a number of major concerns regarding several points listed below, which will require clarification and probably changes that will need to be taken into account.”

Specific comments from R2

2) R2: “Figure 1 is for me misleading; I am not sure that it is correct and relevant to try to obtain a single characterization of the annual cycles (biomass and rate...) for the whole Southern Ocean (from 30S to 60S). Especially with the other figures in the manuscript, which reveal important regional differences. In addition, recent literature has shown how the biogeography, productivity and phenology of the Southern Ocean can be complex and diverse (e.g. Thomalla et al. 2011, Ardyna et al. 2017; 2019, Briggs et al. 2017, Uchida et al. 2019).”

Reply: The value of Figure 1 is that it permits a more direct comparison with previous remote sensing analyses which also provide single characterizations of wide oceanic regions (e.g., Behrenfeld et al., 2017). The identification of similar phytoplankton growth dynamics provides a degree of validation for the use of both remote sensing and in situ float measurements in the study of upper ocean biological and biogeochemical dynamics. We agree with R2 in that the biogeography of the Southern Ocean is diverse, which is why we analyze in further detail the four main zones within this major oceanic region. We have modified the revised manuscript (ms) to point out the value of Figure 1, but also highlight the importance of analyzing each environmental zone in detail: “*Even when integrated over our full Southern Ocean domain, the extensive float record analyzed here immediately highlights the important role of predator-prey relationships in terms of governing the annual phytoplankton biomass cycle. The temporal misalignment between μ and r can also be observed in individual float time series (Figure S3). The synoptic view of the float-based multi-year record (Figure 1) agrees with the broad-scale dynamics of upper ocean planktonic ecosystems inferred from satellite observations for large regions of the high latitude ocean (Behrenfeld et al., 2017). However, the Southern Ocean is comprised of well-established distinct environmental zones that can provide more detailed understanding of biomass variability in this large region of the global ocean (Figure S1). We therefore subdivided the Southern Ocean into four primary zones of differing physical and biogeochemical characteristics*” (line 93–102).

3) R2: “Increasing stratification (decreasing MLD) will not have a linear effect on phytoplankton growth across the Southern Ocean. This is supported by two recent studies by Ardyna et al. 2017 and Llort et al. 2019, which observe a non-monotonic response of phytoplankton biomass to changes in MLD and, more importantly, that a shallowing (or deepening) of the mixed layer is not expected to result in the same bio-

logical response everywhere. The scenario proposed here is really too simplistic. Can you try to get a more comprehensive understanding/overview of the potential changes in growth and NPP in different parts of the Southern Ocean?”

Reply: The studies of Llorc et al. (2019) and Ardyna et al. (2017) focus on the role of bottom-up nutrient (i.e., iron) supply in stimulating phytoplankton biomass. They argue that changes in (primarily) winter mixed layer depths can entrain deep high iron concentrations within the upper mixed layer, affecting the evolution of the phytoplankton bloom. Ardyna et al. (2017) evaluates blooms based on peaks in biomass concentration, without distinguishing between iron effects on phytoplankton division rate (μ , dependent on nutrient and light) vs. net growth rate of phytoplankton biomass (r , dependent of growth as well as loss processes). An issue with the study of Llorc et al. (2019) is that the conclusion of a non-monotonic response between net primary productivity (NPP) and mixed-layer depth (MLD) emerges from the sensitivity experiment where only the depth of the winter mixed layer is varied, but the summer MLD and ferricline remain unchanged. Changes in summer mixed layer could alter the resupply of other important nutrients for phytoplankton groups in this region (i.e., diatoms) such as silicate. Thus, any relationship between changes in MLD, μ , and NPP needs to account for environmental alterations throughout the entire seasonal cycle. Furthermore, increases in iron availability from winter mixing imply a deepening of the mixed layer, which goes against projected changes induced by enhanced ocean stratification.

We avoid assuming any functional form for the relationship between changes in MLD and μ . Our sensitivity analysis is indeed simple, but it is based on the well consented view that ocean stratification will enhance (suppress) μ during winter (summer) at high latitudes (Riebesell et al., 2009; Sarmiento et al., 2004; Schneider et al., 2008). Our analysis benefits from in situ observations of phytoplankton biomass, net biomass rate of change (r), and the estimated decoupling between growth and loss processes (Figure S8), which is likely not included in most model studies. We agree that projected changes in phytoplankton seasonal bloom magnitude will likely be heterogeneous across the Southern Ocean. However, the central message of a decline in bloom magnitude applies to high latitudes in general and it has been reported based on the analysis of satellite ocean color data (Polovina et al., 2008).

We have modified and retitled subsection “Modeling changes in phytoplankton seasonal bloom magnitude” to improve the description of the assessment of future changes in bloom magnitude and net primary production (revised ms, section Methods).

4) R2: “How can we reconcile what is actually observed with a BCG-Argo float, with blooms that generally last less than a month (Briggs et al. 2018, Uchida et al. 2019, Ardyna et al. 2019) with smoothed annual cycles? Can you really translate what you observe based on these smoothed annual cycles to individual times series of the floats (e.g., a concrete example of an annual cycle of a BGC-Argo float could be a relevant addition to support your conclusion)?”

5) R2: “Regarding the smoothed annual cycles, I am concerned about the data processing with three successive smoothing methods (for example the last one covering 60 days, this period is really long especially

in the SIZ), will the results remain similar without the smoothing? This is a critical point that should be addressed and included in the SI.”

Reply: The general seasonality in phytoplankton division rates (μ) and net biomass rate of change (r) can be observed in individual float time series with multi-annual observations (Figure 3 of this reply document, also added as Figure S3 in the revised ms). The temporal misalignment between μ and r is observed in individual float time series, where the seasonal cycle of μ and r becomes more aligned for floats further south, closer to the Seasonal Ice Zone (SIZ), consistent with what is observed in the weekly-averaged seasonal climatologies (Figure 2 and 3 of the revised ms). These features become already evident with a reduced smoothing over a 20 days window. Pulses of biomass increase and accumulation could be observed by analyzing fluctuations in bio-optical properties over shorter time scales. One caveat to consider is that the temporal resolution of most SOCCOM floats is 10 days, which prevents the analysis of short-term events of biological production or export (Briggs et al., 2011).

Successive smoothing is only applied for the multi-annual Southern Ocean time series shown in Figure 1 of the revised ms. The secondary moving filter is applied to reduce small temporal variability that propagates into the computation of the temporal derivatives. We agree with the reviewer that excessive smoothing can result in spurious patterns. The weekly-averaged climatologies for the STZ, SAZ, PAZ, and SIZ consist only of one single smoothing step of a 60 days average-moving window (solid lines in Figure 2 and 3 of the revised ms). The un-smoothed data (i.e., weekly averages) can be observed as individual points in Figures 2 and 3 (revised ms). We have modified section “Methods”, subsection “Time series smoothing” to clarify these steps in the revised ms (please see specific modifications in the submitted document with highlighted changes between the previous and revised versions of the manuscript).

6) R2: “Furthermore, I do not understand why the authors use only 4 years of data for the BGC-Argo floats. The use of all years and all available data will definitely improve the characterization of the different provinces and will even give some indication of intra- regional variability. I also noticed that the old Geotraces 2015 dataset was used, please update it with the more recent Geotraces 2017 dataset (<https://www.bodc.ac.uk/geotraces/data/idp2017/>), to improve Fig S2 and Fig 4b.”

Reply: The analysis of only 4 years of data (2015–2019) only applies to the multi-annual time series for the entire Southern Ocean (Figure 1 of the revised ms). For this synoptic time series, we chose years that include sufficient float profiles in all subregions (STZ, SAZ, PAZ, and SIZ) of the Southern Ocean (south of 30°S). This selection is based on the spatial distribution of floats profiles accumulated in each year where data is available (Figure 4 in this reply letter).

The complete data set (2012–2019) is used in the computation of weekly-averages for the characterization of the environmental zones. We have modified different sections of the revised ms to clarify this feature, particularly in section “Methods”, subsection “Float data”. We have downloaded the “GEO-TRACES_IDP2017_v2_Discrete_Sample_Data.nc”. The documentation of this file is somewhat confusing, and the amount of dissolved iron observations seems to be lower than in the 2015 data set. The distribution

of observations in the present dataset spans from 1978 to 2014 (Figure 5 of this reply letter) (Tagliabue et al., 2012).

Figure 3: Time series of modeled phytoplankton division rates (μ) and observationally-based net phytoplankton biomass rate of change (r) for individual SOCCOM floats (a) f6091, (b) f7620, (c) f9092, and (d) f9091. Grey shaded areas indicate blooming periods ($r > 0$). Time series have been smoothed using a moving average filter over a 20 days window. Light blue and red shaded panels indicate austral winter (May-August) and summer (November-February) months, respectively. Maps show in red the displacement of each float in the Southern Ocean (trajectories in gray are for all float profiles available in the present SOCCOM data set).

Figure 4: Annually accumulated SOCCOM floats profiles for 2012–2019.

Figure 5: Distribution of dissolved iron observations (south of 30°S) of in the dataset obtained from <https://www.bodc.ac.uk/geotraces/data/historical/> (Tagliabue et al., 2012).

References

- Ardyna, M., Claustre, H., Salle, J.-B., D’Ovidio, F., Gentili, B., van Dijken, G., D’Ortenzio, F., and Arrigo, K. R. (2017). Delineating environmental control of phytoplankton biomass and phenology in the southern ocean. *Geophysical Research Letters*, 44(10):5016–5024.
- Arteaga, L., Pahlow, M., and Oschlies, A. (2014). Global patterns of phytoplankton nutrient and light colimitation inferred from an optimality-based model. *Global Biogeochemical Cycles*, 28(7):648–661.
- Banase, K. (1992). Grazing, temporal changes of phytoplankton concentrations, and the microbial loop in the open sea. In Falkowski, P. G., Woodhead, A., and Vivirito, K., editors, *Primary Productivity and Biogeochemical Cycles in the Sea*, volume 43 of *Environmental Science Research*, pages 409 – 440. Springer US, Boston, MA.
- Banase, K. (1994). Grazing and zooplankton production as key controls of phytoplankton production in the open ocean. *Oceanography*, 7:13 – 20.
- Behrenfeld, M. J. (2010). Abandoning Sverdrup’s critical depth hypothesis on phytoplankton blooms. *Ecology*, 91(4):977–989.
- Behrenfeld, M. J. (2014). Climate-mediated dance of the plankton. *Nature Climate Change*, 4:880–887.
- Behrenfeld, M. J., Boss, E., Siegel, D. A., and Shea, D. M. (2005). Carbon-based ocean productivity and phytoplankton physiology from space. *Global Biogeochemical Cycles*, 19:GB1006. doi:10.1029/2004GB002299.
- Behrenfeld, M. J. and Boss, E. S. (2018). Student’s tutorial on bloom hypotheses in the context of phytoplankton annual cycles. *Global Change Biology*, 24(1):55–77.
- Behrenfeld, M. J., Doney, S. C., Lima, I., Boss, E. S., and Siegel, D. A. (2013). Annual cycles of ecological disturbance and recovery underlying the subarctic atlantic spring plankton bloom. *Global Biogeochemical Cycles*, 27(2):526–540.
- Behrenfeld, M. J., Hu, Y., O’Malley, R. T., Boss, E. S., Hostetler, C. A., Siegel, D. A., Sarmiento, J. L., Schullien, J., Hair, J. W., Lu, X., Rodier, S., and Scarino, A. J. (2017). Annual boom-bust cycles of polar phytoplankton biomass revealed by space-based lidar. *Nature Geoscience*, 10:118–122.
- Behrenfeld, M. J. and Milligan, A. J. (2013). Photophysiological expressions of iron stress in phytoplankton. *Annual Review of Marine Science*, 5(1):217–246. PMID: 22881354.
- Boss, E. and Behrenfeld, M. (2010). In situ evaluation of the initiation of the North Atlantic phytoplankton bloom. *Geophysical Research Letters*, 37(18). L18603.
- Boyd, P. W., Jickells, T., Law, C. S., Blain, S., Boyle, E. a., Buesseler, K. O., Coale, K. H., Cullen, J. J., de Baar, H. J. W., Follows, M., Harvey, M., Lancelot, C., Levasseur, M., Owens, N. P. J., Pollard, R., Rivkin, R. B., Sarmiento, J., Schoemann, V., Smetacek, V., Takeda, S., Tsuda, A., Turner, S., and Watson, a. J. (2007). Mesoscale iron enrichment experiments 1993-2005: Synthesis and future directions. *Science*, 315:612–617.

- Briggs, N., Perry, M. J., Cetini, I., Lee, C., D’Asaro, E., Gray, A. M., and Rehm, E. (2011). High-resolution observations of aggregate flux during a sub-polar North Atlantic spring bloom. *Deep Sea Research Part I: Oceanographic Research Papers*, 58(10):1031 – 1039.
- Geider, R. J. (1987). Light and temperature dependence of the carbon to chlorophyll ratio in microalgae and cyanobacteria: Implications for physiology and growth of phytoplankton. *New Phytologist*, 106(1):1–34.
- Geider, R. J. and LaRoche, J. (1994). The role of iron in phytoplankton photosynthesis, and the potential for iron-limitation of primary productivity in the sea. *Photosynthesis Research*, 39:275–301.
- Graff, J. R., Westberry, T. K., Milligan, A. J., Brown, M. B., Dall’Omo, G., van Dongen-Vogels, V., Reifel, K. M., and Behrenfeld, M. J. (2015). Analytical phytoplankton carbon measurements spanning diverse ecosystems. *Deep Sea Research Part I: Oceanographic Research Papers*, 102:16 – 25.
- Johnson, K. S., Plant, J. N., Coletti, L. J., Jannasch, H. W., Sakamoto, C. M., Riser, S. C., Swift, D. D., Williams, N. L., Boss, E., Haëntjens, N., and et al. (2017). Biogeochemical sensor performance in the SOCCOM profiling float array. *Journal of Geophysical Research: Oceans*, 122(8):6416–6436.
- Johnson, K. S., Riser, S. C., Boss, E. S., Talley, L. D., Sarmiento, J. L., Swift, D. D., Plant, J. N., Maurer, T. L., Key, R. M., Williams, N. L., Wanninkhof, R. H., Dickson, A. G., Feely, R. A., and Russell, J. L. (2019). SOCCOM float data — Snapshot 2019-03-12. In *Southern Ocean Carbon and Climate Observations and Modeling (SOCCOM) Float Data Archive*. UC San Diego Library Digital Collections.
- Llort, J., Lvy, M., Salle, J. B., and Tagliabue, A. (2019). Nonmonotonic response of primary production and export to changes in mixed-layer depth in the southern ocean. *Geophysical Research Letters*, 46(6):3368–3377.
- Longhurst, A. (1993). Seasonal cooling and blooming in tropical oceans. *Deep Sea Research Part I: Oceanographic Research Papers*, 40(11):2145 – 2165.
- Mann, K. H. and Lazier, J. R. N. (2006). *Dynamics of Marine Ecosystems. Biological–Physical Interactions in the Oceans*. Blackwell, Boston, Mass.
- Martin, J., Gordon, R. M., and Fitzwater, S. E. (1990). Iron in Antarctic waters. *Nature*, 345:156–158. doi:10.1038/345156a0.
- Mignot, A., Ferrari, R., and Claustre, H. (2018). Floats with bio-optical sensors reveal what processes trigger the North Atlantic bloom. *Nature Communications*, 9(1):190.
- Miller, C. B. (2004). *Biological Oceanography*. Blackwell, Boston, Mass.
- Moore, C. M., Mills, M. M., Arrigo, K. R., Berman-Frank, I., Bopp, L., Boyd, P. W., Galbraith, E. D., Geider, R. J., Guieu, C., Jaccard, S. L., Jickells, T. D., La Roche, J., Lenton, T. M., Mahowald, N. M., Marañón, E., Marinov, I., Moore, J. K., Nakatsuka, T., Oschlies, A., Saito, M. A., Thingstad, T. F., Tsuda, A., and Ulloa, O. (2013). Processes and patterns of oceanic nutrient limitation. *Nature Geoscience*, 6.
- Polovina, J. J., Howell, E. A., and Abecassis, M. (2008). Ocean’s least productive waters are expanding. *Geophysical Research Letters*, 35:L03618.

- Riebesell, U., Körtzinger, A., and Oschlies, A. (2009). Sensitivities of marine carbon fluxes to ocean change. *Proceedings of the National Academy of Sciences*, 106(49):20602–20609.
- Sarmiento, J. L., Slater, R., Barber, R., Bopp, L., Doney, S. C., Hirst, A. C., Kleypas, J., Matear, R., Mikolajewicz, U., Monfray, P., V. S., Spall, S. A., and Stouffer, R. (2004). Response of ocean ecosystems to climate warming. *Global Biogeochemical Cycles*, 18:GB3003. doi:10.1029/2003GB002134.
- Schneider, B., Boop, L., Gehlen, M., Segschneider, J., Frölicher, T. L., Cadule, P., Doney, S. C., Behrenfeld, M. J., and Joos, F. (2008). Climate-induced interannual variability of marine primary and export production in three global coupled climate carbon cycle models. *Biogeosciences*, 5:597–614.
- Siegel, D. A., Doney, S. C., and Yoder, J. A. (2002). The North Atlantic Spring Phytoplankton Bloom and Sverdrup’s Critical Depth Hypothesis. *Science*, 296(5568):730–733.
- Smith, M. J., Tittensor, D. P., Lyutsarev, V., and Murphy, E. (2015). Inferred support for disturbance-recovery hypothesis of north atlantic phytoplankton blooms. *Journal of Geophysical Research: Oceans*, 120(10):7067–7090.
- Tagliabue, A., Mtshali, T., Aumont, O., Bowie, A. R., Klunder, M. B., Roychoudhury, A. N., and Swart, S. (2012). A global compilation of dissolved iron measurements: focus on distributions and processes in the Southern Ocean. *Biogeosciences*, 9(6):2333–2349.
- Westberry, T., Behrenfeld, M. J., Siegel, D. A., and Boss, E. (2008). Carbon-based primary productivity modeling with vertically resolved photoacclimation. *Global Biogeochemical Cycles*, 22:GB2024.
- Westberry, T. K., Schultz, P., Behrenfeld, M. J., Dunne, J. P., Hiscock, M. R., Maritorena, S., Sarmiento, J. L., and Siegel, D. A. (2016). Annual cycles of phytoplankton biomass in the subarctic Atlantic and Pacific Ocean. *Global Biogeochemical Cycles*, 30(2):175–190.
- Yang, B., Boss, E. S., Hantjens, N., Long, M. C., Behrenfeld, M. J., Eveleth, R., and Doney, S. C. (2020). Phytoplankton Phenology in the North Atlantic: Insights from profiling float measurements. *Frontiers in Marine Science*, 7:139.

Reviewers' Comments:

Reviewer #2:

Remarks to the Author:

The authors respond appropriately to my comments and I therefore support the publication of this study.

Just a follow-up comment in response to one of my comments. The study by Polovina et al (2008) was not conducted at high latitudes but at tropical latitudes, and to my knowledge, no study has shown a decline in the bloom magnitude at high latitudes. At least in the Arctic Ocean, the opposite is true.

Mathieu Ardyna

Reviewer #3:

Remarks to the Author:

Review, Arteaga et al 2020

Summary:

Arteaga et al assemble data from drifting floats in the Southern Ocean. Combined with satellite data and model products, they demonstrate a decoupling of phytoplankton division rate from the rate of change in biomass. They show a correlation between bloom magnitude and iron concentration, and conduct a sensitivity test suggesting a disproportionate response in bloom magnitude to changes in division rate.

Overall comment:

Unfortunately I think this article suffers from major problems. The first part is poorly substantiated and does not emphasize any novel contribution, and the second part poses an extremely simplistic (in contrast with simple) speculation as a quantitative conclusion about the potential impacts of climate change. I believe this manuscript needs far more than major revisions before being deemed acceptable for publication, especially in a journal like Nature Communications. There are also additional problematic points that need further explanation and/or analysis before it is clear that they are supported by the evidence: in addition to the major criticisms, see comments indicated with an "X" below. I next explain my conclusion with major criticisms, and then follow with constructive comments as suggestions for a future version of the manuscript. Finally, as requested, I comment on whether they addressed the comments of Reviewer #1.

Major criticisms:

1. The model producing the division rates was not developed for application to the Southern Ocean (which the authors admit, l. 347-351), and I am not convinced that it provides a meaningful estimate of division rates. Specifically, as the authors admit, the use of Chl:C to indicate nutrient limitation was developed for nitrate-limited growth. The authors themselves write "an important caveat of the growth model used here is that it is not clear how well the Chl:C ratio can represent physiological effects of iron limitation on phytoplankton growth." The authors state that growth here was not limited by nitrate, but by iron, as expected for the Southern ocean. Exacerbating this problem is the failure to communicate and illustrate a meaningful uncertainty of the calculated μ in Fig. 1. Since the

difference between μ and r produces the main conclusions of the paper, the conclusions remain poorly substantiated.

2. Even if μ was supported by another line of evidence (and with its uncertainty quantified), the conclusion made -- that biomass accumulation precedes increases in division rates -- is a conclusion that has been made in many previous papers by the authors. Indeed, one of these is a "student's tutorial" on the subject (Behrenfeld and Boss 2018). The new dataset from the floats is fascinating and worth publishing in some form. However, the current article is underwhelming because of this redundancy. Furthermore, the narrative seems to be set up as if the authors have newly discovered the decoupling between μ and r , and then find that it agrees with their other papers in the last section. In my opinion, this is not a sufficiently novel contribution for this journal.

3. The conclusion that a 10% change in division rate may be associated with a 50% reduction in bloom magnitude and NPP is based on a simplistic (vs. simple) model, and is very likely wrong, and because of this, is a meaningless statement (as well as potentially harmful, given the charged politics surrounding climate change). Apart from the fact that the magnitude of growth rate change is not at all mechanistic (as Reviewer #2 pointed out), the same loss rate is assumed for all of the scenarios. In reality, the loss rate is fueled by grazing biomass and thus by the phytoplankton biomass itself, and thus I should track μ to some degree. Thus the big question is to what degree I will track μ . The point of conducting a quantitative sensitivity study is to produce meaningful quantitative estimates. Simple experiments can be useful, and so I understand the authors' motivation to find a simple way to explain a dynamic for such an important topic. However, this is too simplistic of an exercise for publication. This is especially so because of the importance of the topic: inaccurate statements about the impacts of climate change can divert progress in research and beyond. As is, the authors should only qualitatively point out the implication of their analysis which is that small changes in the gross rates can potentially result in large changes in biomass. Note: Reviewer #2 also criticized this exercise as simplistic. Reviewer #1 did not comment on this section at all.

Constructive comments for a future manuscript:

-X- In Fig. 1, $r = 0$ doesn't often line up with the local minima and maxima of the biomass. Specifically, the biomass typically starts to increase in the middle of the winter (the middle of the light blue bars), but $r=0$ at the end of the winter (the edge of the light blue bars). What explains this?

-X- The only deviation from the curves illustrated in Fig. 1 is for PAR, which is not very interesting. Since the differences in μ and r are the main points, their uncertainty should be quantified and illustrated here with error bars or shading.

-X- The relevance of the calculated division rate μ should be demonstrated (perhaps with comparison to a seasonal cycle of growth rate from a biogeochemical model, if not observations?).

-- Fig. 1: The tick marks in Fig. 1 (Jun and Dec marks) don't line up in the same way across the red and blue shaded areas. Why?

-- The most interesting finding of the paper, to me, was the information about phytoplankton dynamics under sea-ice (l. 157-175). I'd suggest turning this into one of the main conclusions. This is much more unique and significant of a contribution compared to some of the other points made in the abstract (such as the correlation with iron.) This point merits more analysis, and so that could easily turn into its own section.

-X- Though the under-ice data was very interesting, I wonder if because the condition for being "under-ice" was that 50% of the data is from under ice (l. 164), the apparent biomass accumulation is mostly contributed by the floats not under the ice. Can you demonstrate that this is not the case?

-- Predator-prey cycles aren't mentioned until l. 94, but they are the mechanism underlying the decoupling. It will be helpful to readers to make this connection earlier. I'd suggest including mention of this mechanism in the introduction.

-- Similarly, the "Disturbance-Recovery" hypothesis isn't addressed until the last section, when it seems as if all of the points made in the first part of the paper are in line with this hypothesis. Introduce this hypothesis earlier, and rather than framing the paper as having come to the same conclusion, as, coincidentally, many of the other Behrenfeld papers, show how your data supports or doesn't support it (in the case of XXX for du/dt).

-- l. 116 (and throughout): When you discuss du/dt , it would be helpful to clarify that this is an empirical correlation (not mechanistic, since it's grazing or other actual loss rates that are the mechanisms). Since in the next section, du/dt does NOT explain the data, you could also just remove this concept from the text entirely. It might end up confusing readers who have been thinking about predator-prey cycles (or other mortality), which are likely the actual mechanisms producing the decoupling.

-- Fig. 1: Remove "rM" from Fig. 1. It is not clear which point along the curve you mean, precisely, and it is easy for the reader to visually locate the minimum of the r curve.

-- l. 104: What differentiates SAZ from PAZ?

-- l. 133 "a direct physical trigger" -- I would say that growth due to nutrient injection is just as much of a physical trigger as decreased grazing rates due to dilution.

-- l. 176: I would suggest cutting this whole section (and Fig. 4). See above critique.

-- l. 209: The "Disturbance-Recovery" hypothesis should be clearly cited -- it is not clear in which previous paper this "new" hypothesis is first stated.

-- l. 399: Do you mean the annual mean was subtracted from the daily values, and not vv ?

-- l. 413: Do you mean the same climatological loss rate was used for all of the scenarios?

Response to comments from Reviewer # 1:

The authors have responded sufficiently to most Reviewer #1's comments. The only major comment not entirely taken into account (part of their major comment 3) was:

"Because of the many caveats in modeling μ , the authors should be careful in how they draw conclusions. For example, it would be best to rephrase their claims as contingent on the (light and chl;c based) model for μ ."

In response, the authors added the caveats of the model in the Methods section, and referred to μ as a model product in the manuscript. But I do not think that they rephrased their claims given the uncertainty in μ (see my above major criticism #1).

Dear editors and reviewers,

Please find below our detailed responses to the comments made by the reviewers on the manuscript titled “Seasonal modulation of phytoplankton biomass in the Southern Ocean”, submitted to *Nature Communications*. We would like to take the opportunity to thank the reviewers for the effort they put into reviewing our manuscript, as well as for the valuable feedback leading to the improvement of the manuscript. Please also find attached a document with highlighted changes between the previous and revised versions of the manuscript.

Reviewer (R) 2

1) R2: “The authors respond appropriately to my comments and I therefore support the publication of this study. Just a follow-up comment in response to one of my comments. The study by Polovina et al (2008) was not conducted at high latitudes but at tropical latitudes, and to my knowledge, no study has shown a decline in the bloom magnitude at high latitudes. At least in the Arctic Ocean, the opposite is true.”

Reply: We agree with this statement, thus, when referencing to the study by Polovina et al (2008) we refer specifically to the Southern Ocean as a region that could present “*flatter seasonal biomass cycles that more closely resemble current lower latitude regions.*” (lines: 224 – 225 of the revised manuscript (RM)).

Reviewer (R) 3

General comments

R3: “Review, Arteaga et al 2020” “Summary: Arteaga et al assemble data from drifting floats in the Southern Ocean. Combined with satellite data and model products, they demonstrate a decoupling of phytoplankton division rate from the rate of change in biomass. They show a correlation between bloom magnitude and iron concentration, and conduct a sensitivity test suggesting a disproportionate response in bloom magnitude to changes in division rate.”

“Overall comment:” “Unfortunately I think this article suffers from major problems. The first part is poorly substantiated and does not emphasize any novel contribution, and the second part poses an extremely simplistic (in contrast with simple) speculation as a quantitative conclusion about the potential impacts of climate change. I believe this manuscript needs far more than major revisions before being deemed acceptable for publication, especially in a journal like Nature Communications. There are also additional

problematic points that need further explanation and/or analysis before it is clear that they are supported by the evidence: in addition to the major criticisms, see further explanation and/or analysis before it is clear that they are supported by the evidence: in addition to the major criticisms, see comments indicated with an "X" below. I next explain my conclusion with major criticisms, and then follow with constructive comments as suggestions for a future version of the manuscript. Finally, as requested, I comment on whether they addressed the comments of Reviewer #1."

R3: "Major criticisms:"

1) R3: "The model producing the division rates was not developed for application to the Southern Ocean (which the authors admit, l. 347- 351), and I am not convinced that it provides a meaningful estimate of division rates. Specifically, as the authors admit, the use of Chl:C to indicate nutrient limitation was developed for nitrate-limited growth. The authors themselves write "an important caveat of the growth model used here is that it is not clear how well the Chl:C ratio can represent physiological effects of iron limitation on phytoplankton growth." The authors state that growth here was not limited by nitrate, but by iron, as expected for the Southern ocean. Exacerbating this problem is the failure to communicate and illustrate a meaningful uncertainty of the calculated μ in Fig. 1. Since the difference between μ and r produces the main conclusions of the paper, the conclusions remain poorly substantiated."

Reply: We have conducted several new analyses to validate and provide confidence in our float-based estimates of division rates (μ) obtained from the CbPM. In these analyses, we find that division rates estimated from the CbPM compare well with a productivity algorithm parameterized specifically for Southern Ocean waters (Arrigo2008, Arrigo et al., 2008). Furthermore, we compare both models with division rates estimates inferred from a data base of in situ carbon-14 (^{14}C) based net primary productivity measurements (Buitenhuis et al., 2013). We show that float-based division rates estimates from the CbPM compare well and more favorably with Southern Ocean in situ-based μ than estimates from the Southern Ocean oriented algorithm (Arrigo2008). Our results indicate that the CbPM is able to reasonably predict phytoplankton division rates (μ) in the Southern Ocean, thereby, providing a robust validation constraint for the results presented in our study.

We have included an entire new section in the Supplementary Information (SI) of the revised manuscript (RM) describing this new assessment of division rates from the CbPM in the Southern Ocean: Section **Assessment of division rates estimated by the CbPM** (lines 39–96 of the SI, and presented below). We highlight this analysis to the reader in the initial section of the main text of the RM: "*The validity of modeled division rates was assessed by comparing them with a productivity algorithm parameterized specifically for Southern Ocean waters (Arrigo et al., 2008) as well as division rate estimates inferred from in situ carbon-14 (^{14}C) based net primary productivity measurements (Buitenhuis et al., 2013) (Supplementary Information).*" (lines 82–85). We also refer the reader to this analysis in the section Methods, subsection "Phytoplankton growth model": "*Division rates obtained by the CbPM in the Southern Ocean are similar to outputs of μ obtained from a productivity algorithm parameterized specifically for Southern Ocean waters (Arrigo et al., 2008), and compare favorably with estimates of μ derived from in situ carbon-14 (^{14}C) based net primary*

productivity measurements (Buitenhuis et al., 2013) (Supplementary Information).” (lines 376–379 of the RM).

Assessment of division rates estimated by the CbPM

Division rate (μ) outputs from the CbPM were compared against a productivity algorithm parameterized specifically for Southern Ocean waters (Arrigo2008, Arrigo et al., 2008). The Arrigo2008 algorithm computes division rates as:

$$\mu(z, t) = G_{\max}(t) \times L(z, t). \quad (1)$$

$$G_{\max}(t) = G_0 e^{rT(t)} \quad (2)$$

where, following the Arrigo2008 (Equation 8 and 9) notation (Arrigo et al., 2008), μ at a given time (t) and depth (z) depends on $G_{\max}(t)$, the temperature (T) dependent upper limit to net phytoplankton growth rate (i.e., division rate) and an irradiance limitation term (L). G_0 is the phytoplankton net growth rate at 0°C (0.59 d^{-1}) and r is a rate constant ($0.0633 \text{ }^\circ\text{C}^{-1}$) that determines the sensitivity of $G_{\max}(t)$ to temperature. The light limitation term, $L(z, t)$, is calculated for each depth and each time step as:

$$L(z, t) = 1 - e^{-\frac{\text{PUR}(z, t)}{E'_k(z, t)}} \quad (3)$$

where PUR is the photosynthetically usable radiation (assumed equivalent to PAR) and E'_k is the spectral photoacclimation parameter (see Equations 10–14 of the Arrigo2008 algorithm description for details on the computation of these parameters: Arrigo et al., 2008).

The temporal evolution of division rate estimated in our study by the CbPM agrees well with that estimated from the Southern Ocean-aimed formulation of Arrigo2008 (Figure 1a). In particular, the ‘timing’ (increase/decrease) of both estimates of μ follows a similar seasonal cycle over the interannual time series of profiling floats observations, providing confidence in the in situ-based observation of a temporal lag between division rates and the net biomass rate of change (r) (Figure 1 of the main RM).

The CbPM allows for a decomposition of the nutrient and light controlling effects on μ . Nutrient limitation (low nutrient index) is diagnosed to occur during summer months, in opposite fashion to the annual cycle of the light index (Figure 1b). The impact of nutrient stress on μ is relatively low (i.e., the nutrient index only decreases to about 0.6) which might be due to not explicitly accounting for iron limitation in the model. However, the timing of low nutrient index in summer and high in winter is consistent with the seasonal expectation of micronutrient availability in the Southern Ocean (Tagliabue et al., 2014) and therefore provides confidence in the ability of the model to detect the correct seasonality of nutrient limitation in this region. As observed above, a productivity algorithm parameterized specifically for Southern Ocean waters (Arrigo2008, Arrigo et al., 2008) presents a very similar seasonality in μ .

We compare float-based estimates of division rates obtained from the CbPM and Arrigo2008 with division rates estimated from a data base of in situ carbon-14 (^{14}C) net primary productivity measurements (Buitenhuis et al., 2013). In order to infer division rates from in situ-based measurements of vertically integrated NPP, we computed $\mu = \frac{\text{NPP}_{\text{int}}}{C_{\text{phyto}} \cdot Z_{\text{eu}}}$ where NPP_{int} ($\text{mg C m}^2 \text{ d}^{-1}$) is vertically integrated ^{14}C -based net primary production over the euphotic depth, C_{phyto} (mg C m^3) is the mean phytoplankton concentration in the euphotic depth, and Z_{eu} (m) is the euphotic depth. C_{phyto} and Z_{eu} were obtained from a monthly-resolved climatological merged product of MODIS and CALIOP satellite observations used to infer NPP and marine carbon export (Arteaga et al., 2018). In situ-based estimates of μ are obtained by matching the monthly-resolved satellite-based climatologies with the same month at which ^{14}C NPP measurements were obtained. The in situ-based estimates of division rate show a coherent global pattern, with high μ in the equatorial Pacific Ocean and at high latitudes, and low μ in subtropical regions (Figure 2c). Due to reduced in situ observations in the Southern Ocean, we subsample float-based estimates of μ by averaging all float-based estimates within a horizontal radius of 500 km around each in situ observation at the same month of the year. This results in a total of 187 comparable data points south of 30°S .

A strong correlation between float-based and in situ-based estimates of division rates was not initially expected, since ^{14}C NPP measurements were combined with satellite biomass data to infer μ , float and in situ data are not spatially coincident, and both data sets were monthly matched but do not coincide perfectly in time (they represent different years). Despite these sources of discrepancy between data sets, we find that float-based division rates estimated from the CbPM compare well and more favorably with in situ-based μ ($R^2 = 0.25$, root-mean-square error of the scatter around the least-squares linear fit ($\text{RMSE}_{\text{fit}} = 0.24 \text{ d}^{-1}$, root-mean-square error between linear fit and one-to-one line ($\text{RMSE}_{\text{model}} = 0.35 \text{ d}^{-1}$) (Figure 2a) than estimates from Arrigo2008 ($R^2 = 0.13$, $\text{RMSE}_{\text{fit}} = 0.19 \text{ d}^{-1}$, $\text{RMSE}_{\text{model}} = 0.58 \text{ d}^{-1}$) (Figure 2b). These results indicate that the CbPM is able to reasonably predict phytoplankton division rates (μ) in the Southern Ocean, thereby, providing a robust validation constraint for the results presented in this study.

2) R3: “Even if μ was supported by another line of evidence (and with its uncertainty quantified), the conclusion made – that biomass accumulation precedes increases in division rates – is a conclusion that has been made in many previous papers by the authors. Indeed, one of these is a “student’s tutorial” on the subject (Behrenfeld and Boss 2018). The new dataset from the floats is fascinating and worth publishing in some form. However, the current article is underwhelming because of this redundancy. Furthermore, the narrative seems to be set up as if the authors have newly discovered the decoupling between μ and r , and then find that it agrees with their other papers in the last section. In my opinion, this is not a sufficiently novel contribution for this journal.”

Reply: The observation that biomass accumulation precedes division rates and its implication for establishing what drives seasonal phytoplankton blooms has so far primarily been based on remote sensing satellite observations. The Disturbance-Recover hypothesis (DRH) is still not a well established hypothesis partly due to the lack of large scale in situ-based evidence and analyses to corroborate satellite observations. The lack of consensus regarding the DRH has been evident during the review process of this paper, where R1 suggested that the theory was speculative to a certain degree, while R3 seems to suggest that this finding is

Figure 1: (a) Annual cycles of float-based mean phytoplankton division rates in the upper mixed layer computed from the CbPM (black-continuous line) and the Arrigo2008 (Arrigo et al., 2008) algorithm (black-dashed line). (a) Annual cycles of the mean mixed layer light (red-continuous line) and nutrient (blue-continuous line) saturation indices of phytoplankton growth inferred from the CbPM.

well established and therefore not sufficiently novel. The explicit omission of the DRH until the last section of the paper was a modification performed on the original version of the manuscript to address some of the comments made by R1.

The verification of satellite-based patterns with in situ data is critical for the validation and wide acceptance of any ecological theory, including the DRH. Our study provides clear evidence that ecological mechanisms primarily inferred from synoptic satellite analyses are also observed in large scale in situ observations of the Southern Ocean. Furthermore, by focusing on the Southern Ocean, our results provide evidence that the general mechanisms of the DRH are applicable in diverse regions of the global ocean, different from the well studied North Atlantic spring bloom, for which the DRH was initially developed (Behrenfeld, 2010). We agree with R3 in that the DRH should be introduced earlier in the manuscript. We now introduce the DRH in the introductory section of the revised manuscript to provide early context of the results presented in our study (lines 55–59): *“These ‘bloom-forming’ mechanisms have been previously summarized by the ‘Disturbance-Recovery’ hypothesis based predominantly on satellite observations (Behrenfeld, 2010; Behrenfeld et al., 2017). Our results provide the first large scale observational evidence in support of this hypothesis, based primarily on in situ biogeochemical and bio-optical data from autonomous drifting floats”.*

Figure 2: (a) Scatter plot of in situ-based and float-based estimates of upper ocean phytoplankton division rates inferred from the (a) CbPM ($R^2 = 0.25$, $RMSE_{\text{fit}} = 0.24 \text{ d}^{-1}$, $RMSE_{\text{model}} = 0.35 \text{ d}^{-1}$) and (b) the Arrigo2008 algorithm (Arrigo et al., 2008) ($R^2 = 0.13$, $RMSE_{\text{fit}} = 0.19 \text{ d}^{-1}$, $RMSE_{\text{model}} = 0.58 \text{ d}^{-1}$). Solid black-continuous line is the output from the linear regression model. Black dashed-line represents the one-to-one line. (c) Global patterns of phytoplankton division rates estimated from in situ carbon-14 (^{14}C) based net primary productivity measurements (Buitenhuis et al., 2013) used to validate float based estimates of μ . Modeled outputs of μ in the Southern Ocean were subsampled by averaging all float-based estimates within a horizontal radius of 500 km around each in situ observation at the same month of the year. This resulted in a total of 187 comparable data points south of 30°S shown in the scatter plots (a and b).

3) R3: “The conclusion that a 10% change in division rate may be associated with a 50% reduction in bloom magnitude and NPP is based on a simplistic (vs. simple) model, and is very likely wrong, and because of this, is a meaningless statement (as well as potentially harmful, given the charged politics surrounding climate change). Apart from the fact that the magnitude of growth rate change is not at all mechanistic (as Reviewer #2 pointed out), the same loss rate is assumed for all of the scenarios. In reality, the loss rate is fueled by grazing biomass and thus by the phytoplankton biomass itself, and thus I should track μ to some degree. Thus the big question is to what degree I will track μ . The point of conducting a quantitative sensitivity study is to produce meaningful quantitative estimates. Simple experiments can be useful, and so I understand the authors’ motivation to find a simple way to explain a dynamic for such an important topic. However, this is too simplistic of an exercise for publication. This is especially so because of the importance of the topic: inaccurate statements about the impacts of climate change can divert progress in research and beyond. As is, the authors should only qualitatively point out the implication of their analysis which is that small changes in the gross rates can potentially result in large changes in biomass. Note: Reviewer #2 also criticized this exercise as simplistic. Reviewer #1 did not comment on this section at all.”

Reply: Our evaluation of the general sensitivity of the Southern Ocean phytoplankton bloom magnitude and NPP to changes in μ is rooted on expected seasonal changes in the availability of nutrient and light as a result of projected upper ocean stratification at high latitudes. While the actual quantification of changes in NPP and bloom magnitude is based on a simplified average estimate for the entire Southern Ocean, our estimates of division (μ), loss (l), net biomass rate of change (r), and NPP are based on in situ observations, providing a robustness and constraint that cannot be achieved by the large degree of uncertainty embedded in coupled circulation-biogeochemical models. Presently, it is unlikely that any given circulation-biogeochemical model can accurately predict the impact that upper ocean stratification will have on future phytoplankton division rates. Therefore, we rely on the mechanistic assumption that changes in upper ocean stratification can derive on a wide range of variations in μ (from 10% to 60%), and establish a change in seasonality in accordance with expected changes in nutrient and light (e.g., increased/decreased μ in winter/summer, respectively) (Sarmiento et al., 2004; Riebesell et al., 2009).

Contrary to what expressed by R3, the same loss rate is not assumed for all scenarios. The loss rate is recalculated at each sensitivity run based on the resulting seasonal cycle of μ , where l lags μ by two days (see Figure S8 of the RM as well as Behrenfeld and Boss (2018)). This is now clarified in lines 202-205 of the RM: “*For these simulations, we assumed that loss rates paralleled changes in μ but with a temporal lag (Behrenfeld and Boss, 2018) (Methods). The loss rate is recalculated at each sensitivity run based on the resulting seasonal cycle of μ .*”. We do understand the concern of R3 in reporting drastic changes in marine ecosystems that could be misinterpreted under the context of climate change. Therefore, we have modified the title of this section to “*Controls and sensitivity of phytoplankton seasonal bloom in the Southern Ocean*” (line 186 of RM). We also improve the explanation and interpretation of our sensitivity analysis in lines 212–225 of the RM: “*These results indicate that a relatively small change of 10% in μ can result in a relatively large (estimated here at $\sim 50\%$) reduction in bloom magnitude and NPP. The highly sensitive response of the bloom magnitude and NPP to a reduction in summer μ results from the fact that division rates decrease during the period of highest (exponential) phytoplankton growth. These estimates are mostly*

theoretical and are obtained for the average seasonal cycle in phytoplankton biomass in the Southern Ocean, but they respond to the expected general trend in nutrient and light availability if upper ocean stratification is to increase at high latitudes (Sarmiento et al., 2004; Riebesell et al., 2009). The exact quantitative reduction in bloom magnitude and NPP will, nevertheless, depend on the impact that reduced surface ocean mixing will have on division and loss rates, as well as on the compound net environmental change of the heterogeneous Southern Ocean. While the consequences of changes in NPP on oceanic carbon export and sequestration remains to be quantified, our analysis suggests that relatively small changes in phytoplankton division rates in the Southern Ocean could result in flatter seasonal biomass cycles that more closely resemble current lower latitude regions (Polovina et al., 2008).“

Constructive comments for a future manuscript from R3

4) R3: “-X- In Fig. 1, $r = 0$ doesn’t often line up with the local minima and maxima of the biomass. Specifically, the biomass typically starts to increase in the middle of the winter (the middle of the light blue bars), but $r=0$ at at end of the winter (the edge of the light blue bars). What explains this?”

Reply: As described in the Methods section, subsection “Net biomass rate of change”, r is computed following Equation 4 of the RM, which describes a “switching algorithm where r is computed from changes in phytoplankton concentration during periods of mixed layer shoaling and from changes in phytoplankton inventory during periods of mixed layer deepening (or stationary)” (lines 311–314). Therefore, r does not perfectly align with either integrated or average upper ocean phytoplankton biomass. We still think it is important to show the interannual time series of integrated biomass (Figure 1a, RM) to provide context for the seasonality, magnitude, and variability of the biomass stock in the Southern Ocean.

Additionally, the Supplementary Information of the manuscript (lines 25 –38) details the difference between the computation of r during periods of mixed layer shoaling or deepening, and the impact that the algorithm has on the final computation of r (Figure S6 and S7).

5) R3: “-X- The only deviation from the curves illustrated in Fig. 1 is for PAR, which is not very interesting. Since the differences in μ and r are the main points, their uncertainty should be quantified and illustrated here with error bars or shading.”

Reply: The variability in r and μ is primarily driven by the variation in biomass which is represented by the individual blue dots around the mean biomass time series (black line) in Figure 1a of the RM. The addition of uncertainty lines in Figure 1c of the RM impaires the visualization of the temporal lag between r and μ . Therefore, we had added seasonally-resolved frequency distributions of r , μ , and the main float-based phytoplankton variables computed in our study in Figure S2 of the Supplementary Information.

6) R3: “-X- The relevance of the calculated division rate μ should be demonstrated (perhaps with comparison to a seasonal cycle of growth rate from a biogeochemical model, if not observations?).”

Reply: We have addressed this issue by comparing our float-based μ with division rate estimates derived from in situ ^{14}C based net primary productivity measurements (see Reply 1).

7) R3: “– Fig. 1: The tick marks in Fig. 1 (Jun and Dec marks) don’t line up in the same way across the red and blue shaded areas. Why?”

Reply: We thank the reviewer for noting this. Initially the x-tick marks were located each at 180 days from the start date of the time series, slightly displacing the date at which the mark was displayed in each year. We have now fixed this to display x-tick marks on January/01 (summer) and July/01 (winter).

8) R3: “– The most interesting finding of the paper, to me, was the information about phytoplankton dynamics under sea-ice (l. 157-175). I’d suggest turning this into one of the main conclusions. This is much more unique and significant of a contribution compared to some of the other points made in the abstract (such as the correlation with iron.) This point merits more analysis, and so that could easily turn into its own section.”

Reply: We now highlight our under sea-ice results in abstract of the RM: “*Under ice observations show that biomass starts increasing in early winter, well before sea ice begins to retreat.*”(lines 25–26). Results from under sea-ice floats are framed within its own section in the “Subtropical and Seasonal Ice Zones”. We have chosen to describe under ice floats results here to keep consistency with the overall structure of the manuscript and its length requirements. Further analysis of under ice dynamics will be detailed in upcoming manuscripts by researchers from the SOCCOM project.

9) R3: “-X- Though the under-ice data was very interesting, I wonder if because the condition for being “under-ice” was that 50% of the data is from under ice (l. 164), the apparent biomass accumulation is mostly contributed by the floats not under the ice. Can you demonstrate that this is not the case?”

Reply: This is a good point. The condition to define the under ice period is that *at least* 50% of floats profiles obtained during that period proceed from under ice floats. However, the fraction of under-ice float profiles during the defined “under ice” period is often $> 70\%$ (Figure 3a of this reply document). Therefore, most of the r data during this period represents indeed under ice dynamics. We recomputed the net biomass rate of change for the SIZ including *only* under ice observations during the under-ice period (Figure 3a of this reply document). The observed seasonality in r is essentially indistinguishable from that of Figure 3b of the RM (division rates cannot be obtained when only under ice profiles are analyzed).

10) R3: “– Predator-prey cycles aren’t mentioned until l. 94, but they are the mechanism underlying the decoupling. It will be helpful to readers to make this connection earlier. I’d suggest including mention of this mechanism in the introduction.”

Reply: Good suggestion. As mention in Reply 2, we now introduce the Disturbance-Recovery hypothesis in the introductory section of the RM to provide context of the results presented in our study and to highlight early in the manuscript the central role of predator-prey cycles (lines 55–59).

a)

b)

Figure 3: (a) Percentage of SOCCOM float profiles under ice during each week of year (black line) (blue line represents period where at least 50% of profiles are under ice). (b) Annual cycle of net phytoplankton biomass rate of change (r , blue line) and division rates (μ , red dots) for the SIZ, including only under-ice floats during the under-ice period (blue shaded area).

11) R3: “– Similarly, the ”Disturbance-Recovery” hypothesis isn’t addressed until the last section, when it seems as if all of the points made in the first part of the paper are in line with this hypothesis. Introduce this hypothesis earlier, and rather than framing the paper as having come to the same conclusion, as, coincidentally, many of the other Behrenfeld papers, show how your data supports or doesn’t support it (in the case of XXX for du/dt).”

Reply: This comment is in opposition to what initially suggested by reviewer 1. As mentioned above, this further highlights the discrepancy in the community regarding the acceptance of the DRH, and the importance of validating the DRH with in situ data (as presented in our study) to support the patterns observed and ecological processes inferred from space. As indicated in Reply 2 and 10, we agree with R3 in that the DRH should be introduced earlier in the manuscript, therefore, it is now introduced in lines 55–59 of the RM: “These ‘bloom-forming’ mechanisms have been previously summarized by the ‘Disturbance-Recovery’

hypothesis based predominantly on satellite observations (Behrenfeld, 2010; Behrenfeld et al., 2017). Our results provide the first large scale observational evidence in support of this hypothesis, based primarily on in situ biogeochemical and bio-optical data from autonomous drifting floats”.

12) R3: “– l. 116 (and throughout): When you discuss du/dt , it would be helpful to clarify that this is an empirical correlation (not mechanistic, since it’s grazing or other actual loss rates that are the mechanisms). Since in the next section, du/dt does NOT explain the data, you could also just remove this concept from the text entirely. It might end up confusing readers who have been thinking about predator- prey cycles (or other mortality), which are likely the actual mechanisms producing the decoupling.”

Reply: We are not sure that we understand this comment. Changes in $d\mu/dt$ are not considered to just empirically correlate with r , but to drive the net biomass rate of change through accelerations/decelerations in μ . This acceleration (deceleration) in division rates allows for a decoupling (recoupling) of growth and grazing rates, affecting the net accumulation of biomass (r). This is evident in the SAZ and PAZ, and it is also identified as the main pattern in lidar satellite observations of the Southern Ocean (i.e., Behrenfeld et al., 2017) because the SAZ and PAZ occupy the largest area of the Southern Ocean. Nevertheless, we certainly do not want to confuse the readers regarding how changes in division rate are linked to changes in grazing, so in the revised text we highlight the relevance of changes in $d\mu/dt$ controlling r because it is the main mechanism described in annual cycles of polar phytoplankton biomass previously revealed by space-based lidar (Behrenfeld et al., 2017), and we would like to make sure this link is clear. However, while the effects of accelerations and decelerations in μ on grazing were dominant over large sections of the Southern Ocean, in the STZ and SIZ we surprisingly found that other factors seem to be more important in decoupling/recoupling growth and grazing rates.

13) R3: “– Fig. 1: Remove ”rM” from Fig. 1. It is not clear which point along the curve you mean, precisely, and it is easy for the reader to visually locate the minimum of the r curve.”

Reply: rM is an important event of the annual cycle of biomass because it marks the moment when r stops decreasing but is still < 0 (lines 95–97 of the RM). This shows that the net biomass rate of change can start increasing (become less negative) even during conditions of declining division rates. This highlights that loss processes must play a role in controlling the annual cycle of biomass, and therefore, we believe that it is important to indicate this point of the annual cycle.

14) R3: “– l. 104: What differentiates SAZ from PAZ?”

Reply: As stated in section “Methods”, subsection “Environmental zones”: “*Environmental zones defined in the Southern Ocean (Orsi et al., 1995) are based on a mean 2004–2014 Argo-based climatology of temperature and salinity (Roemmich and Gilson, 2009; Bushinsky et al., 2017) (Figure S1).*”. These zones have been defined to keep consistency with other studies within the SOCCOM project. Specific details can be found in Bushinsky et al. (2017):

- *The Subantarctic Zone (SAZ, area = $1.96 \times 10^{13} \text{ m}^2$) lies between the Subtropical Front and the*

Subantarctic Front (location where θ at 400 dbar equals 5°C) and is characterized by deep wintertime mixed layer depths, reaching a mean of 280 ± 140 m in September, before rising to 48 ± 21 m in January. Nitrate is never fully drawn down in this region, with minimum float measured values observed in January of $8.3 \pm 3.1 \mu\text{mol kg}^{-1}$ and maximum values of $16 \pm 3.1 \mu\text{mol kg}^{-1}$ occurring in August, during maximum exchange with deep waters. The amplitude of the seasonal nitrate change is greater in this region than in any of the others.

- *The Polar Frontal-Antarctic Zone (PAZ, area = 2.69×10^{13} m²) extends from the Subantarctic Front to the September extent of sea ice (15% concentration) for the 2014/2015 winter. Mean winter MLDs in the PAZ are 130 ± 69 m, similar to the STZ, while summertime MLDs average 48 ± 26 m. The seasonal nitrate drawdown is ~ 4 to $5 \mu\text{mol kg}^{-1}$, with minimum values measured from March to June, and a maximum in September.*

15) R3: “– l. 133 ”a direct physical trigger” – I would say that growth due to nutrient injection is just as much of a physical trigger as decreased grazing rates due to dilution.”

Reply: Agreed. We have changed this phrase to: “*A direct physical trigger on grazing rates for the SAZ autumn blooms may be the primary driver of this event...*” (lines 143 –144 of the RM).

16) R3: “– l. 176: I would suggest cutting this whole section (and Fig. 4). See above critique.”

Reply: Please see Reply 3.

17) R3: “– l. 209: The ”Disturbance-Recovery” hypothesis should be clearly cited – it is not clear in which previous paper this ”new” hypothesis is first stated.”

Reply: The DRH originated from satellite observations of the Subarctic Atlantic presented in Behrenfeld (2010), and has continued to evolve, with most recent observations relevant to our work in Behrenfeld et al. (2017). We have added these two references in line 230 of the RM: “*A new ‘Disturbance-Recovery’ hypothesis has been proposed that accommodates these findings (Behrenfeld, 2010; Behrenfeld et al., 2017)...*”.

18) R3: “– l. 399: Do you mean the annual mean was subtracted from the daily values, and not vv?”

Reply: We are not sure what “vv” means. As stated in the RM (lines 425–426): “*Seasonal anomalies in μ are calculated by subtracting the climatological daily value of μ from the overall annual mean of μ .*”

19) R3: “– l. 413: Do you mean the same climatological loss rate was used for all of the scenarios?”

Reply: We thank the reviewer for alerting us to this potential point of confusion. We now clarify in the RM that: “*The loss rate is recalculated at each sensitivity run based on the resulting seasonal cycle of μ .*” (lines 204–205).

R3: Response to comments from Reviewer # 1:

20) R3: “ The authors have responded sufficiently to most Reviewer #1’s comments. The only major comment not entirely taken into account (part of their major comment 3) was: ”Because of the many caveats in modeling μ , the authors should be careful in how they draw conclusions. For example, it would be best to rephrase their claims as contingent on the (light and chl;c based) model for μ .”

In response, the authors added the caveats of the model in the Methods section, and referred to μ as a model product in the manuscript. But I do not think that they rephrased their claims given the uncertainty in μ (see my above major criticism #1) ”.

Reply: We have addressed the validity of our float-based division rates by comparing them with a productivity algorithm parameterized specifically for Southern Ocean waters (Arrigo2008, Arrigo et al., 2008), and with in situ-base μ estimates derived from ^{14}C net primary productivity measurements and satellite-based biomass. The seasonality of μ is highly similar in the CbPM and Arrigo2008 model, providing confidence in our in situ-based observation of a temporal lag between division and net biomass rate of change (μ and r , respectively). Moreover, estimates of μ from our study (CbPM) compare well and more favorably with in situ-based division rates than those obtained from the Southern Ocean productivity algorithm (Arrigo2008). Please see Reply 1 to reviewer 3 and Figures 1 and 2 of this reply document for details. This analysis has been included in the Supplementary information of the RM: Section **Assessment of division rates estimated by the CbPM**.

References

- Arrigo, K. R., van Dijken, G. L., and Bushinsky, S. (2008). Primary production in the Southern Ocean, 19972006. *Journal of Geophysical Research: Oceans*, 113(C8). C08004.
- Arteaga, L., Haeëntjens, N., Boss, E., Johnson, K. S., and Sarmiento, J. L. (2018). Assessment of export efficiency equations in the Southern Ocean applied to satellite-based net primary production. *Journal of Geophysical Research: Oceans*.
- Behrenfeld, M. J. (2010). Abandoning Sverdrup’s critical depth hypothesis on phytoplankton blooms. *Ecology*, 91(4):977–989.
- Behrenfeld, M. J. and Boss, E. S. (2018). Student’s tutorial on bloom hypotheses in the context of phytoplankton annual cycles. *Global Change Biology*, 24(1):55–77.
- Behrenfeld, M. J., Hu, Y., O’Malley, R. T., Boss, E. S., Hostetler, C. A., Siegel, D. A., Sarmiento, J. L., Schullien, J., Hair, J. W., Lu, X., Rodier, S., and Scarino, A. J. (2017). Annual boom-bust cycles of polar phytoplankton biomass revealed by space-based lidar. *Nature Geoscience*, 10:118–122.
- Buitenhuis, E. T., Hashioka, T., and Qur, C. L. (2013). Combined constraints on global ocean primary production using observations and models. *Global Biogeochemical Cycles*, 27(3):847–858.
- Bushinsky, S. M., Gray, A. R., Johnson, K. S., and Sarmiento, J. L. (2017). Oxygen in the Southern Ocean from argo floats: Determination of processes driving air-sea fluxes. *Journal of Geophysical Research: Oceans*, 122(11):8661–8682.
- Orsi, A. H., Whitworth, T., and Nowlin, W. D. (1995). On the meridional extent and fronts of the Antarctic Circumpolar Current. *Deep Sea Research Part I: Oceanographic Research Papers*, 42(5):641 – 673.
- Polovina, J. J., Howell, E. A., and Abecassis, M. (2008). Ocean’s least productive waters are expanding. *Geophysical Research Letters*, 35:L03618.
- Riebesell, U., Körtzinger, A., and Oschlies, A. (2009). Sensitivities of marine carbon fluxes to ocean change. *Proceedings of the National Academy of Sciences*, 106(49):20602–20609.
- Roemmich, D. and Gilson, J. (2009). The 2004–2008 mean and annual cycle of temperature, salinity, and steric height in the global ocean from the argo program. *Progress in Oceanography*, 82(2):81 – 100.
- Sarmiento, J. L., Slater, R., Barber, R., Bopp, L., Doney, S. C., Hirst, A. C., Kleypas, J., Matear, R., Mikolajewicz, U., Monfray, P., V., S., Spall, S. A., and Stouffer, R. (2004). Response of ocean ecosystems to climate warming. *Global Biogeochemical Cycles*, 18:GB3003. doi:10.1029/2003GB002134.
- Tagliabue, A., Salle, J.-B., Bowie, A. R., Lvy, M., Swart, S., and Boyd, P. W. (2014). Surface-water iron supplies in the Southern Ocean sustained by deep winter mixing. *Nature Geoscience*, 7(4):314–320.

Reviewers' Comments:

Reviewer #3:

Remarks to the Author:

The authors have sufficiently responded to most of my comments. Particularly, they have provided justification for their model of the division rates in the Southern Ocean with extensive additional analysis. The discussion of the sensitivity study has been modified to better communicate that the results should not be taken literally as a projection of future oceans. The mention of the Disturbance-Recovery Hypothesis in the introduction better prepares the readers for the results and specifies the novel contribution of the manuscript. The manuscript has been much improved on account of the substantial revisions from three reviewers.

One point remains: The lack of uncertainty/error in the mean μ and r curves in Fig. 1. I understand that Fig. 1 is busy and that adding more information to the plot while retaining clarity is challenging. However, again, the lag between μ and r leads to the main conclusions of the paper, and quantifying and communicating the uncertainty in these mean values is important to support these conclusions. There are certainly many examples in the literature of comparative time series with error bars or shaded areas of uncertainty. It is not clear how the spread of the blue dots in Fig. 1a relates to the uncertainty in μ or r . The distributions in the Supplement do provide some of this information, but as seasonal averages only. It doesn't seem too much to ask for the blue and red lines in Fig. 1c to include a lightly shaded region (as a suggestion) indicating the uncertainty in the mean values (standard error, or, if Gaussian curves are not expected, the error from the 5th and 95th percentiles). The grey shading in 1c could be removed if necessary, and blue and red shaded areas added, as one specific suggestion of how to make this change. For another, the color indicating the seasons is not that necessary given the words "Winter" and "Summer" on the top axis, so that could be changed to grey shades, or indicated with lines (dotted for winter?) in order to reduce the total number of colors in the final figure.

Minor follow-up: In my previous comment #18 (previous l. 399), "vv" was shorthand for "vice versa" - - apologies for the shorthand. To further explain, I would have expected that the mean was subtracted from the daily values, such that a positive daily value is converted into a positive anomaly. I.e., the current sentence "subtracting the ... daily value ... from the .. mean" implies:

anomaly = mean - daily value

whereas shouldn't it be:

anomaly = daily value - mean

Dear editors and reviewers,

Please find below our detailed responses to the remaining concerns of the reviewer on the manuscript titled “Seasonal modulation of phytoplankton biomass in the Southern Ocean”, submitted to *Nature Communications*. Once again, we would like to take the opportunity to thank the reviewers for the effort they put into reviewing our manuscript, as well as for the valuable feedback leading to the improvement of the manuscript. Please also find attached a document with highlighted changes between the previous and revised versions of the manuscript.

Reviewer (R) 3

Remarks to the Author

“The authors have sufficiently responded to most of my comments. Particularly, they have provided justification for their model of the division rates in the Southern Ocean with extensive additional analysis. The discussion of the sensitivity study has been modified to better communicate that the results should not be taken literally as a projection of future oceans. The mention of the Disturbance-Recovery Hypothesis in the introduction better prepares the readers for the results and specifies the novel contribution of the manuscript. The manuscript has been much improved on account of the substantial revisions from three reviewers.”

1) R3: “One point remains: The lack of uncertainty/error in the mean μ and r curves in Fig. 1. I understand that Fig. 1 is busy and that adding more information to the plot while retaining clarity is challenging. However, again, the lag between μ and r leads to the main conclusions of the paper, and quantifying and communicating the uncertainty in these mean values is important to support these conclusions. There are certainly many examples in the literature of comparative time series with error bars or shaded areas of uncertainty. It is not clear how the spread of the blue dots in Fig. 1a relates to the uncertainty in μ or r . The distributions in the Supplement do provide some of this information, but as seasonal averages only. It doesn’t seem too much to ask for the blue and red lines in Fig. 1c to include a lightly shaded region (as a suggestion) indicating the uncertainty in the mean values (standard error, or, if Gaussian curves are not expected, the error from the 5th and 95th percentiles). The grey shading in 1c could be removed if necessary, and blue and red shaded areas added, as one specific suggestion of how to make this change. For another, the color indicating the seasons is not that necessary given the words “Winter” and “Summer” on the top axis, so that could be changed to grey shades, or indicated with lines (dotted for winter?) in order to reduce the total number of colors in the final figure.”

Reply: Thank you for the suggestion. We have tried multiple ways of adding shaded areas representing the uncertainty around the multi-annual time series of μ and r in Figure 1c. Given the large amount of colors and labels, particularly in panel c, this results in a very convoluted and confusing figure. Therefore, we have now replicated and simplified Figure 1c and added it to the supplementary material of the revised manuscript (Figure S3), displaying the multi-annual time series of μ and r (similarly as in Figure 1c) with uncertainties represented as the standard deviation of the time series of μ and r (red and blue shaded areas, respectively) (also included as Figure 1 in this reply document).

2) “R3: Minor follow-up: In my previous comment #18 (previous l. 399), ”vv” was shorthand for ”vice versa” – apologies for the shorthand. To further explain, I would have expected that the mean was subtracted from the daily values, such that a positive daily value is converted into a positive anomaly. I.e., the current sentence ”subtracting the ... daily value ... from the .. mean” implies:

anomaly = mean - daily value

whereas shouldn't it be:

anomaly = daily value - mean

?”

Reply: Thank you for the clarification. The interpretation of the reviewer is correct: The mean is subtracted from the daily values, such that a positive daily value is converted into a positive anomaly. We have now corrected and clarified this in line 431 of the revised manuscript:

Seasonal anomalies in μ are calculated by subtracting the overall annual mean of μ from the climatological daily value of μ ($\mu_{daily} - \mu_{mean}$).

Figure 1: Average time series of modeled phytoplankton division rates (μ) (red continuous line) and phytoplankton net biomass rate of change rate (r) (blue continuous line) with uncertainties represented as the standard deviation of the time series of μ and r (red and blue shaded areas, respectively) (Same as Figure 1c with displayed uncertainties).